# *Mycobacterium tuberculosis* triggers reduced inflammatory cytokine responses and virulence in mice lacking Tax1bp1

Jeffrey Chin[1], Nalin Abeydeera[1], Teresa Repasy[2¤a], Rafael Rivera-Lugo[2¤b],
Gabriel Mitchell[2], Vinh Q Nguyen[3,4,5,6], Weihao Zheng[7], Alicia Richards[8,9,10],
Erica Stevenson[8,9,10], Danielle L Swaney[8,9,10], Nevan J Krogan[8,9,10], Joel D Ernst[7],
Jeffery S Cox[2], Jonathan M Budzik [1,2¤c]*

**1** Department of Medicine, University of California, San Francisco, California, United States of America,
**2** Department of Molecular and Cell Biology, University of California, Berkeley, California, United States
of America, **3** Gladstone-UCSF Institute of Genomic Immunology, San Francisco, California, United
States of America, **4** Department of Surgery, University of California, San Francisco, California, United
States of America, **5** Diabetes Center, University of California, San Francisco, California, United States of
America, **6** UCSF CoLabs, University of California, San Francisco, California, United States of America,
**7** Division of Experimental Medicine, Department of Medicine, University of California, San Francisco,
California, United States of America, **8** Quantitative Biosciences Institute (QBI), University of California,
San Francisco, San Francisco, California, United States of America, **9** J. David Gladstone Institutes,
San Francisco, California, United States of America, **10** Department of Bioengineering and Therapeutic
Sciences, University of California, San Francisco, California, United States of America

¤a Current address: Seattle Children's Hospital, Seattle, Washington, United States of America
¤b Current address: Department of Biology, Stanford University, Stanford, California, United States of
America
¤c Current address: Department of Medicine, University of California, San Francisco, California, United
States of America
* Jonathan.Budzik2@ucsf.edu

org/10.1371/journal.ppat.1012829

Recherche Scientifique, FRANCE

**Peer Review History:** PLOS recognizes the
benefits of transparency in the peer review
process; therefore, we enable the publication
of all of the content of peer review and
author responses alongside final, published
articles. The editorial history of this article is
available here: https://doi.org/10.1371/journal.
ppat.1012829

## Abstract

Host responses – autophagy, cell death, and inflammation – limit the growth of bacterial pathogens while minimizing tissue damage. During the early stages of infection, *Mycobacterium tuberculosis* (*Mtb*) thwarts these and other innate immune defense mechanisms in alveolar macrophages (AMs) derived from the yolk sac; in later stages, it circumvents defenses in recruited mononuclear cells (MNCs) and survives within them despite additional cytokine stimulation from recruited T cells. The mechanisms that drive variable rates of *Mtb* growth in different macrophage subtypes and how *Mtb* manipulates inflammatory responses to grow within innate immune cells remain obscure. Here we explored the role of the host factor, Tax-1 binding protein 1 (Tax1bp1), an autophagy receptor that targets pathogens for degradation through selective autophagy and terminates pro-inflammatory cytokine responses. Unexpectedly, we found that Tax1bp1-deficient mice were less susceptible to *Mtb* infection, and generated reduced inflammatory cytokine responses, compared to wild-type mice; the same mutant mice exhibited decreased growth of, and inflammatory cytokine responses to, *Listeria monocytogenes*, suggesting that Tax1bp1 plays a role in

**Data availability statement:** Confocal microscopy images, flow cytometry files, and histopathology slides are available on the Dryad repository (DOI: 10.5061/dryad.44j0zpcq6). RNA sequencing files, including the differential gene expression analysis, can be accessed on GEO, accession GSE280399. Mass spectrometry data files are available on ProteomeXChange with accession # PXD064244. The full list of protein abundance changes is available on Dryad.

**Funding:** This work was supported by National Institutes of Health grants K08 AI146267 (JB), R01AI194696 (JB), U19 AI135990 (NJK, JSC), P01 AI063302 (JSC), U19 AI106754 (JSC), DP1 AI124619 (JSC), and R01 AI120694 (JSC). JMB was also supported by the UCSF Nina Ireland Program in the Health Award, Cystic Fibrosis Foundation Harry Shwachman Award (CA-0140443), Mentored Scientist in Tuberculosis Award (R25AI147375), and TB RAMP program (R25AI147375). R.R.-L was supported by a grant from the National Academies of Sciences, Engineering, and Medicine (Ford Foundation Fellowship) and the University of California Dissertation-Year Fellowship. The funders had no role in the study, data collection and analysis, decision to publish, or preparation of the manuscript. Funding URLs: https://www.niaid.nih.gov/ https://www.cff.org/researchers/harry-shwachman-clinical-investigator-award https://tb.ucsf.edu/tb-ramp-tuberculosis-research-and-mentoring-program https://pulmonary.ucsf.edu/ireland https://rap.ucsf.edu/mentored-scientist-award-tuberculosis https://tb.ucsf.edu/tb-ramp-tuberculosis-research-and-mentoring-program https://www.nationalacademies.org/our-work/ford-foundation-fellowships https://gradapp.berkeley.edu/portal/fellowships?cmd=ucdiss.

**Competing interests:** The authors have declared that no competing interests exist.

host responses to multiple intracellular pathogens. Contrary to our previous *ex vivo* findings in bone marrow-derived macrophages (BMDMs), *in vivo* growth of *Mtb* in AMs and a subset of recruited MNCs was more limited in mice lacking Tax1bp1 relative to wild-type mice. To better understand these differences, we performed global protein abundance measurements in mock- and *Mtb*-infected AM samples *ex vivo* from wild-type mice. These experiments revealed that Tax1bp1 protein abundance does not significantly change early after infection in AMs but does in BMDMs; moreover, early after infection, Tax1bp1-deficiency reduced necrotic-like cell death -- an outcome that favors *Mtb* replication -- in AMs but not BMDMs. Together, these results show that deficiency of Tax1bp1 plays a crucial, cell type-specific role in linking the regulation of autophagy, cell death, and anti-inflammatory host responses and overall reducing bacterial growth.

## Author summary

Macrophages are the first innate immune cells to encounter and be infected by *Mycobacterium tuberculosis (Mtb)* during infection. There are at least 9 different types of macrophages, and recent studies suggest that some permit *M. tuberculosis* replication and survival more than others, but the mechanisms for cell type-specific differences in *M. tuberculosis* growth are only beginning to be understood. We found that deficiency of the host factor, Tax1bp1 (Tax-1 binding protein 1), restricts *M. tuberculosis* growth during animal infection and in specific subsets of innate immune cells, including alveolar macrophages, while enhancing *M. tuberculosis* in bone marrow-derived macrophages. We also found that Tax1bp1-deficiency has a similar phenotype in abrogating the pathogenesis of another intracellular pathogen, *Listeria monocytogenes.* During infection, Tax1bp1-deficiency reduced inflammatory cytokine production and neutrophil and CD8[+] T cell recruitment to the lungs of *Mtb*-infected mice. Compared to bone marrow-derived macrophages, in alveolar macrophages, Tax1bp1-deficiency decreased the release of inflammatory mediators and necrotic-like host cell death, a mode of host cell death known to enhance *M. tuberculosis* growth. Tax1bp1 protein level did not change significantly during *Mtb* infection of AMs but did increase significantly during BMDM infection, highlighting a potential mechanism that explains the different responses mediated by Tax1bp1 in BMDMs and AMs. Our research highlights that Tax1bp1 is a host target for host-directed therapy against *M. tuberculosis* and controls host responses to *M. tuberculosis* in a cell type-specific manner.

## Introduction

*Mycobacterium tuberculosis* (*Mtb*), the causative agent of tuberculosis, has evolved the ability to circumvent host innate antimicrobial responses and survive within our

immune cells [1, 2]. In the lung of non-human primates infected with *Mtb*, nine macrophage subtypes were identified [3]. Alveolar macrophages (AMs) are the first immune cells to become infected after inhalation of *Mtb* and provide a replicative niche for intracellular bacteria [4–6]. To disseminate to other organs, *Mtb* must spread from AMs to different cell types, such as recruited monocyte-derived macrophages (MNCs) [4]. In the mouse model of infection, this dissemination occurs at approximately 14–25 days post-infection upon recruitment of monocytes and their differentiation into macrophages in the lung parenchyma [7]. At this same time point, T cells are recruited to the *Mtb*-infected lungs and contribute to cytokine production [8]. Ultimately, live bacteria are transported from the lungs via lymphatics to the draining lymph nodes [7].

In *Mtb*-infected murine lungs, monocytes differentiate into two subsets that differ in their ability to control *Mtb* [9]. The first subset is the CD11c$^{lo}$ subset (MNC1, mononuclear cell subset 1), also known as recruited macrophages [10–12]. The second subset is the CD11c$^{hi}$ subset (MNC2), formerly known as dendritic cells but considered more closely related to macrophages [7, 11–13]. AMs, which are embryonically derived from the yolk sac, and monocyte-derived macrophages originating from circulating monocytes of bone marrow origin [14] exhibit divergent transcriptional responses against *Mtb* [6]. AMs are impaired at mounting antibacterial responses against *Mtb,* which leads to significant *Mtb* growth differences when compared to murine bone marrow-derived macrophages (BMDMs) *ex vivo* [6] or monocyte-derived macrophages *in vivo* [5]. However, the host factors mediating different rates of *Mtb* growth in macrophage subtypes are only beginning to be understood.

Understanding the host factors that limit immune responses to *Mtb* is critical to develop new host-directed antimicrobial therapies [15, 16]. In various macrophage subtypes, immune responses to *Mtb* are generated by the detection of *Mtb* lipoproteins and lipoglycans by macrophage surface receptors (*e.g.,* TLR2 [17, 18], Dectin 1 [19]), which mediate ERK (extracellular-signal regulated kinase) and NF-κB inflammatory signaling [20]. Cytosolic sensors also recognize *Mtb* nucleic acids released during phagosomal perforation [21] and engage several downstream signaling pathways, each of which can have opposing impacts on *Mtb* growth. For example, cytosolic sensing of *Mtb* DNA by cGAS (cyclic GMP–AMP synthase) triggers type I interferon (IFN) production and autophagy [22]. Autophagy is a protective response against *Mtb* that targets it for ubiquitylation and degradation in the lysosome [23–26]. Conversely, type I IFN is a cytokine that protects against viruses but can be co-opted by *Mtb* to promote its growth [27–29]. Another important cytosolic sensor for *Mtb* is the RIG (Retinoic acid-inducible gene I)-I-like pathway. Like the cGAS pathway, the RIG-I-like pathway's cytosolic sensing of *Mtb* RNA induces host responses with opposing effects on *Mtb* growth. RIG-I induces apoptosis, a mode of cell death that leads to *Mtb* growth restriction, while also triggering pro-bacterial type I IFN and limiting NF-κB cytokine production such as TNF-α, IL-1β, and IL-6. The impacts of NF-κB-regulated cytokines can be pro- or anti-bacterial [30, 31]. For instance, TNF-α can restrict *Mtb* by activating phagocytes but, in excess, can enhance *Mtb* growth by mediating tissue damage [32–35]. These *Mtb*-mediated signaling pathways are regulated by post-translational modifications through cascade signaling protein phosphorylation [21, 23, 26, 36]. Thus, *Mtb* infection triggers post-translationally regulated host responses that can have pro- or anti-bacterial effects. Nevertheless, we lack a complete understanding of host factors that drive these responses to thwart *Mtb* growth.

Tax1bp1 is an autophagy receptor at the nexus of multiple key immune responses critical for pathogen control. Tax1bp1 regulates inflammation and blocks apoptosis in response to cytokine stimulation, vesicular stomatitis virus (VSV), and Sendai virus infections by terminating NF-κB and RIG-I signaling [37–43]. In infected Tax1bp1-deficient mice, respiratory syncytial virus (RSV) replication is decreased, whereas cytokine responses are enhanced [38]. The anti-inflammatory function of Tax1bp1 has also been shown to impact non-infectious diseases through abrogating the development of chemically-induced hepatocellular cancer [39], age-dependent dermatitis, and cardiac valvulitis [37]. Tax1bp1 promotes selective autophagy that mediates lysosomal degradation of pathogens, such as *Mtb* in BMDMs [44] and *Salmonella enterica* Typhimurium [45]. Additionally, Tax1bp1 mediates the clearance of aggregated neuronal proteins involved in neurodegenerative disease [46]. Thus, Tax1bp1 has an anti-inflammatory function in several contexts, but the impact of Tax1bp1 *in vivo* during intracellular bacterial infection has hitherto not been described.

Our previous work showed that the autophagy receptor, Tax1bp1, was phosphorylated and its protein abundance increased during *Mtb* infection of BMDMs [44]. Tax1bp1-deficiency increased *Mtb* growth during *ex vivo* infection of BMDMs, presumably because of the role of Tax1bp1 in antibacterial autophagy [44]. Notably, Tax1bp1-deficiency did not significantly change the levels of NF-κB-regulated cytokines during *Mtb* infection of BMDMs [44]. To expand our understanding of Tax1bp1's function in other critical innate immune cell types and the presence of the complete immune system, here we employed the mouse infection models for *Mtb* and the intracellular pathogen *Listeria monocytogenes*. Surprisingly, this led to the discovery that Tax1bp1-deficiency reduced *Mtb* and *Listeria* growth during animal infection and decreased CD8+ T cell and neutrophil recruitment to the *Mtb*-infected lungs. Tax1bp1-deficiency impaired *Mtb* growth in AMs, in contrast to the role of Tax1bp1-deficiency in the enhancement of *Mtb* growth in BMDMs. Furthermore, Tax1bp1-deficiency had an anti-inflammatory function during *Mtb* and *Listeria* animal infection, compared to its pro-inflammatory phenotype in viral infections. A mechanistic difference between *Mtb*-infected AMs and BMDMs that may contribute to these phenotypes is that Tax1bp1 protein abundance increased in BMDMs but not AMs following infection. We found that Tax1bp1-deficiency reduced necrotic-like cell death and inflammatory mediator release during AM but not BMDM infection, which is a likely mechanism by which Tax1bp1 leads to cell type-specific changes in *Mtb* growth. To our knowledge, we are the first to study the function of Tax1bp1 in the context of mouse infections with these pathogens. Our findings from the animal infection model led us to uncover new relevant phenotypes compared to those revealed by the BMDM infection model and revealed Tax1bp1's negative impact on immunity to intracellular pathogens.

## Results

### *In vivo*

**The absence of Tax1bp1 reduces *Mtb* growth during acute and chronic stages of animal infection.** To assess *Tax1bp1's* contribution to controlling *Mtb* infection *in vivo* we exposed wild-type and Tax1bp1-deficient mice to aerosolized, virulent *Mtb.* We first measured colony forming units (CFU) one day after exposing male and female mice to a low dose. We saw no statistically significant difference in bacterial uptake between wild-type and mutant mice of the same sex (Fig 1A). However, female *Tax1bp1-/-* mice had decreased bacterial uptake compared to male *Tax1bp1-/-* mice (Fig 1A). To control for this variable, subsequent animal infections were performed with sex-matched wild-type and *Tax1bp1-/-* mice.

To examine the role of Tax1bp1 in controlling *Mtb* growth, we harvested the lung, spleen, and liver of wild-type and *Tax1bp1-/-* mice on days 9 or 11, 21, and 50 post-infection and quantified bacterial CFUs (Figs 1B and S1). Contrary to our previous results in BMDMs infected *ex vivo* [45], we found that *Mtb* growth decreased in the lungs of Tax1bp1-deficient mice (Figs 1B and S1). This unexpected difference manifests even after 11 days of infection, during the acute stage when *Mtb* replicates within AMs. *Mtb*'s restricted growth was magnified in both peripheral organs, liver, and spleen, consistent with the differences in lung CFUs. These results contrast with the autophagy receptor function of Tax1bp1 since Tax1bp1 contributes to targeting *Mtb* to selective autophagy [46], a host response that can contribute to the control of *Mtb* growth *in vivo* and *ex vivo* to a variable degree [24,47–55].

**Tax1bp1-deficiency abrogates pro-inflammatory cytokine production during *Mtb* infection *in vivo*.** We reasoned, given Tax1bp1's known role in inflammation, that differences in *Mtb* growth between wild-type and Tax1bp11-deficient mice might reflect differences in inflammatory responses to *Mtb* infection. Indeed, consistent with this idea, the lungs of Tax1bp1-deficient mice exhibited decreased levels of IL-6, TNF-α, IL1-β, and IL-12/IL-23 p40 (Figs 1C and S1).

Type I and II IFNs are particularly important for controlling *Mtb* infection [22, 29, 47]. Therefore, we measured interferon levels using a type I and II IFN reporter cell line (ISRE) and type II IFN (IFN-γ) by ELISA (Fig 1C and 1D). Consistent with other pro-inflammatory cytokines, we found that Tax1bp1-deficient mice had significantly lower levels of type I and II IFN in the lungs compared to wild-type mice (Fig 1C and 1D).

Although Tax1bp1-deficiency decreased inflammatory cytokine synthesis during *Mtb* infection, microscopic examination of infected lung tissue did not reveal any significant differences in the cellular infiltrate of the lungs as reflected by lesion

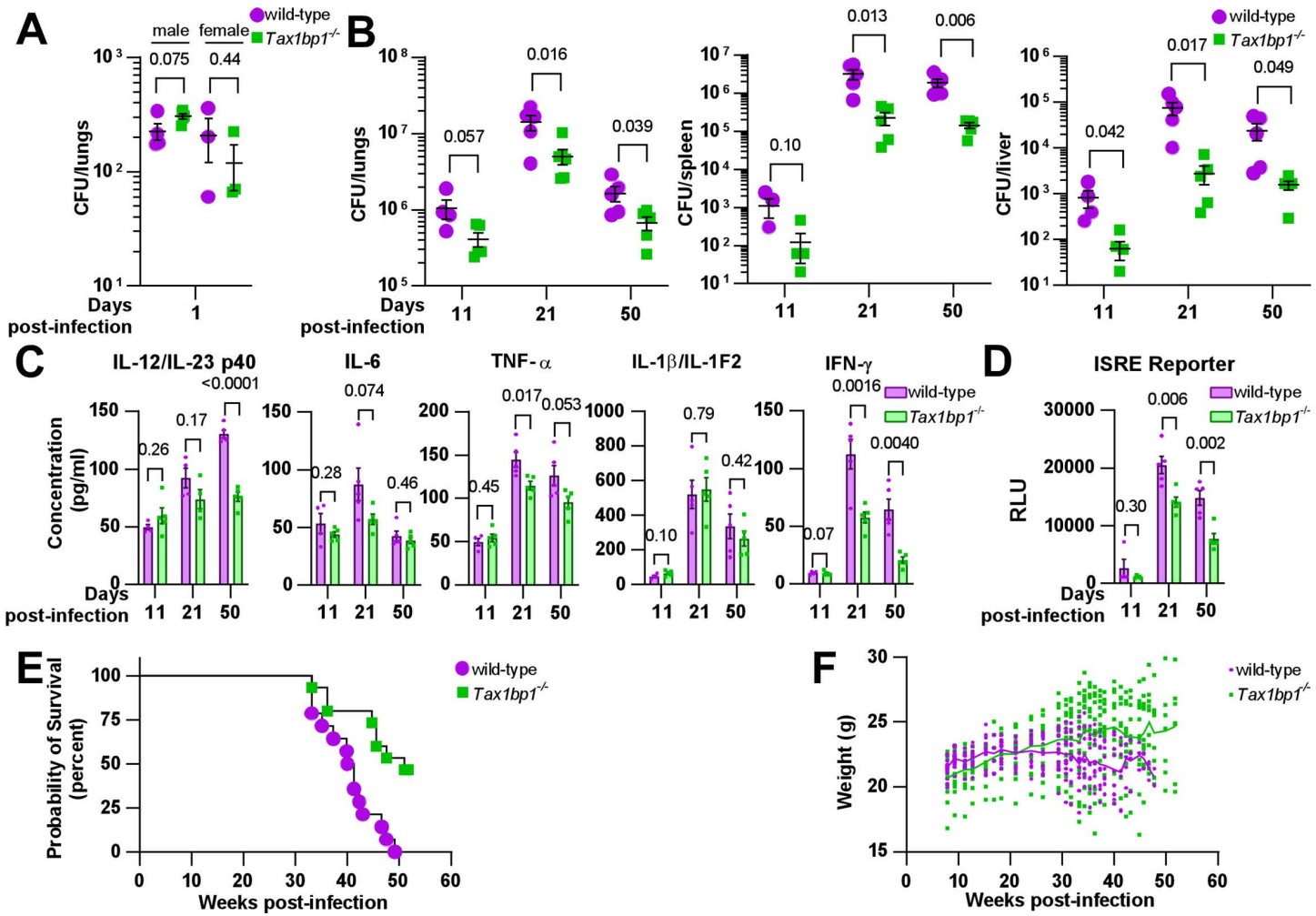

**Fig 1. Tax1bp1-deficiency abrogates *M. tuberculosis* virulence and inflammatory cytokine responses during mouse aerosol infection. (A)** Age and sex-matched male and female mice were infected by the aerosol route with a mean *Mtb* CFU of 240 as determined by CFU enumeration from lung homogenates at 1-day post-infection. **(B)** Male mice were euthanized at 11 and 21-days post-infection for CFU enumeration. Female mice were euthanized at 50-days post-infection. Results are the mean ± SEM from lung homogenates of 4-5 infected mice per genotype. **(C)** Cytokine levels from infected lung homogenates at 11-, 21-, and 50-days post-infection were measured by ELISA. Results are the mean ± SEM from five samples. **(D)** Levels of type I and II interferon-induced JAK/STAT signaling were measured by luminescence in relative light units (RLUs) from infected lung homogenates by the ISRE assay. Results are the mean ± SEM from five samples. Brackets indicate p values from t-test comparisons. **(E)** Infected mice were monitored for death or 15% loss of maximum body weight, at which point they were euthanized. Log-rank (Mantel-Cox) test and Gehan-Breslow-Wilcoxon comparison test p values for survival were 0.0008 and 0.0047, respectively. **(F)** The weight of the infected mice is displayed. The connecting line denotes the mean weight of the mice alive at each time point.

severity or tissue necrosis (S2 Fig). There was a trend that was not statistically significant towards decreased neutrophil recruitment in Tax1bp1-deficient lung samples as reflected by myeloperoxidase staining (S2 Fig).

During infection of BMDMs, we previously observed that Tax1bp1-deficiency enhanced ubiquitin colocalization with *Mtb* [44]. Increased ubiquitylation was also observed in the brains of *Tax1bp1⁻ᐟ⁻* mice [46]. Thus, we sought to determine whether ubiquitin recruitment was regulated during infection *in vivo*. While Tax1bp1 led to a slight decrease in ubiquitin and *Mtb* colocalization in infected lung tissue samples at 50 days post-infection, this did not reach statistical significance (S3 Fig).

Finally, we carried out survival studies and found that Tax1bp1-deficiency contributes to reduced mortality during *Mtb* infection (Fig 1E) and enhances weight gain (Fig 1F). Collectively, our results suggest that Tax1bp1-deficiency limits host-detrimental systemic inflammatory responses and reduces host susceptibility and mortality.

**Tax1bp1-deficiency reduces *Listeria monocytogenes* growth, microabscess formation, and host inflammatory cytokine synthesis**

Next, we sought to assess the role of Tax1bp1 in systemic cytokine responses and bacterial growth in isolation from its role in autophagy and to test whether Tax1bp1 might play a similar role in response to a different intracellular pathogen. Therefore, we exposed Tax1bp1-deficient mice and their macrophages to a pathogen, *Listeria monocytogenes,* that employs mechanisms to inhibit antibacterial autophagy [52–54]. As shown in Fig 2A and 2B, *Listeria* grew as well in both BMDMs and peritoneal macrophages harvested from wild-type and *Tax1bp1*⁻ᐟ⁻ mice, indicating there is no defect in the ability of *Listeria* to replicate in *Tax1bp1*⁻ᐟ⁻ macrophages *ex vivo*. In contrast, when we infected mice with *Listeria* via the intravenous (IV) route in two independent experiments, we saw that over a 48-hour time course, *Tax1bp1*⁻ᐟ⁻ mice were remarkably resistant to *Listeria* growth relative to wild-type animals (Figs 2C and S4). Consistent with Tax1bp1 playing a role early in the *Listeria* infection, we observed a statistically significant difference in *Listeria* CFU in the spleen, but not yet in the liver, at 4 hours post-infection (Fig 2D).

To determine whether Tax1bp1-deficient mice infected with *Listeria* also exhibit reduced inflammatory cytokine responses as we previously observed during *Mtb* infection, we measured inflammatory cytokine levels in the serum of the IV-infected mice and found that during the first 4- and 10 hours, inflammatory cytokines were low and were not significantly different between the wild-type and *Tax1bp1*⁻ᐟ⁻ mice (Fig 2E). At 48 hours, Tax1bp1-deficiency significantly reduced IL-6, TNF-α, IFN-γ, IFN-β and MCP-1 in the serum, indicating Tax1bp1-deficiency abrogates pro-inflammatory cytokine and type I interferon (IFN-β) production (Fig 2E).

Infecting mice by the intraperitoneal route gave rise to very similar results, indicating that the route of infection is irrelevant to the infection outcome (Fig 2F). Indeed, Tax1bp1-deficiency decreased CFU by approximately 1.5 logs in the liver and by 1 log in the spleen (Fig 2F). As with the IV infection, Tax1bp1-deficiency led to a considerable non-statistically significant decrease in IL-6 production and a substantial decrease in IFN-γ in the serum (Fig 2G). Likewise, Tax1bp1-deficiency reduced MCP-1 levels in the serum (Fig 2G).

Consistent with the decreased bacterial load in *Tax1bp1*⁻ᐟ⁻ mice, histological examination of tissues from Tax1bp1-deficient mice exhibited decreased microabscesses and lymphoid depletion (S5 Fig). Tax1bp1-deficiency increased the occurrence and severity of hepatocyte coagulative necrosis (S5 Fig), which indicates hypoxic cell death. The lack of splenic microabscesses and lymphoid depletion of *Tax1bp1*⁻ᐟ⁻ mice may correlate with the decrease in pro-inflammatory cytokine levels noted in the serum of *Tax1bp1*⁻ᐟ⁻ mice compared to wild-type mice. These results suggest that Tax1bp1-deficiency reduces inflammatory cytokine signaling and bacterial growth during animal infection with *Listeria monocytogenes*, as observed during *Mtb* infection.

**Tax1bp1-deficiency impairs the recruitment of neutrophils and CD8⁺ T cells to the lungs during *Mtb* infection.** Having shown that Tax1bp1-deficiency restricts bacterial growth and inflammatory cytokine production during *Mtb* and *Listeria monocytogenes* animal infection, we next sought to further explore how Tax1bp1-deficiency impacts recruitment of inflammatory immune cells during *Mtb* infection. First, we turned to a more sensitive approach compared to histopathology to examine immune cell populations in the lungs. To precisely quantify immune cell types in the lungs during infection with green-fluorescent *Mtb*, we performed analytical flow cytometry of lung homogenates (Fig 3). We quantified two sets of immune cells after gating for live single cells: (1) myeloid cell subsets including CD11c^lo (MNC1) and CD11c^hi (MNC2) mononuclear cells, neutrophils, and AMs (S6 Fig); and (2) B, T, and NK cells (S7 Fig). Using this more sensitive approach revealed (i) decreased neutrophil recruitment and (ii) increased proportions of live AMs at 28 days post-infection in mice lacking Tax1bp1 (Fig 3). One possible explanation for this result is that Tax1bp1-deficiency increased the survival of AMs during *Mtb* infection.

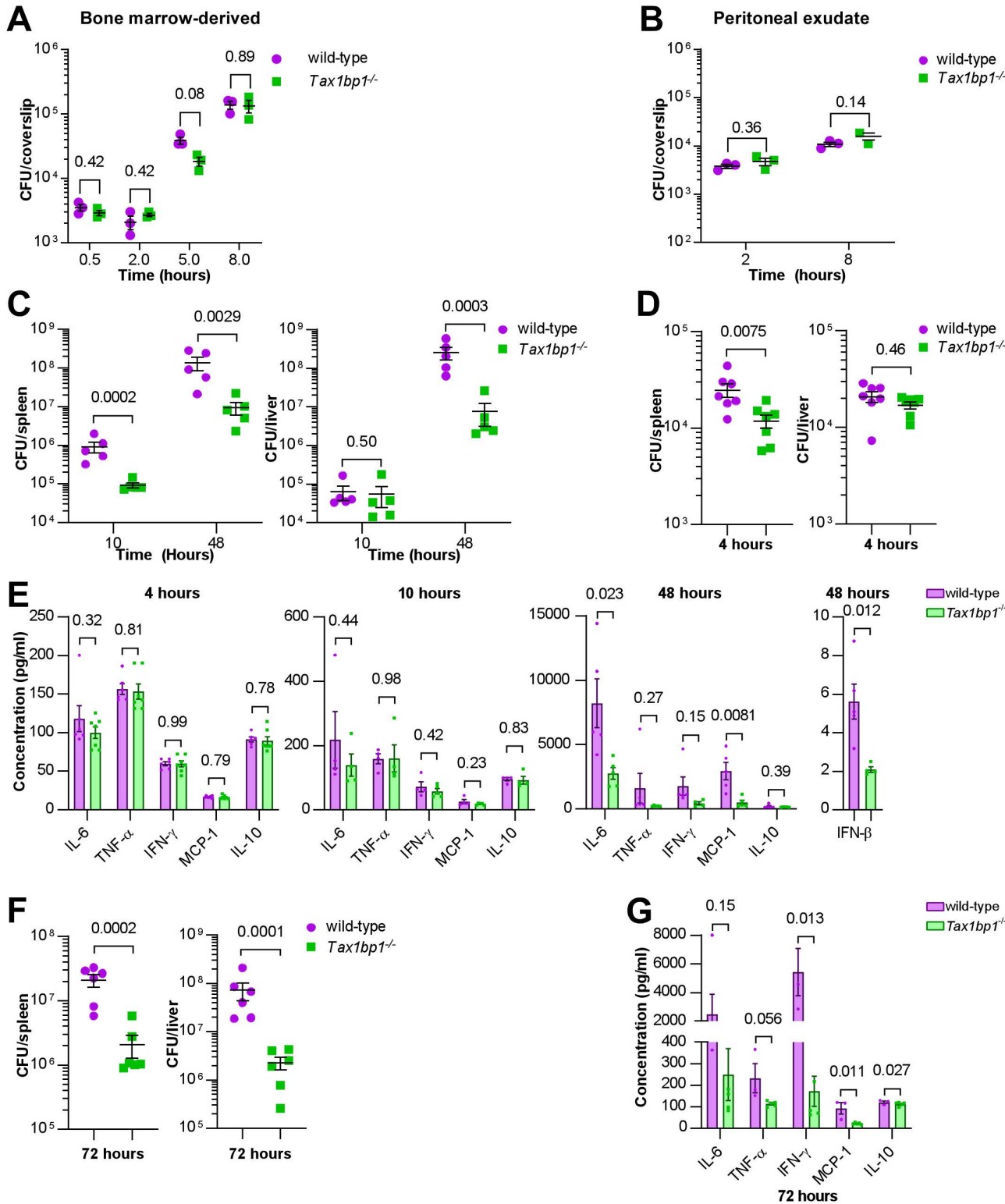

**Fig 2. Tax1bp1-deficiency impaired *Listeria monocytogenes* virulence and growth during murine but not *ex vivo* cellular infections. (A and B)** BMDMs or peritoneal exudate cells were infected with *L. monocytogenes,* and CFU were counted at 30 minutes, 2-, 5-, or 8 hours post-infection. Results are the mean±SEM from three technical replicate samples. The p values from t-test comparisons are shown. **(C and D)** CFU from spleen and

liver homogenates from mice intravenously infected with *L. monocytogenes* were enumerated at 4-, 10-, or 48-hours post-infection. Results are the mean±SEM from five mice. Brackets indicate p values from t-test comparisons. **(E & G)** Cytokine levels were measured from the serum of mice infected with *L. monocytogenes* at 4-, 10-, and 48-hours post-infection by cytometric bead array (IL-6, TNF-α, IFN-γ, MCP-1, IL-10) or ELISA (IFN-β). Results are mean±SEM from five samples. Brackets indicate p values from t-test comparisons. **(F)** Spleen and liver homogenates from mice intraperitoneally infected with *L. monocytogenes* were enumerated for CFU 72 hours post-infection. Results are the mean±SEM from five mice. Brackets indicate p values from t-test comparisons. CFU data were logarithmically transformed prior to statistical analysis.

In addition to decreasing neutrophil recruitment, Tax1bp1-deficiency increased MNC1 recruitment at 28 days post-infection but had no impact on MNC2 recruitment (Fig 3C). Tax1bp1-deficiency reduced CD8+ T cell recruitment by 2 and 2.5-fold at 21- and 28-days post-infection, respectively (Fig 3A). This result is consistent with Tax1bp1-deficiency limiting systemic inflammatory cytokine responses, as CD8+ T cells are associated with the production of cytokines including IFN-γ [57] and TNF-α [58]. In contrast, CD4+ T cell, NK cell, and B cell levels in the lungs were not affected by Tax1bp 1-deficiency (Fig 3A). Together with the decreased cytokine production and lack of pulmonary histopathologic changes mediated by Tax1bp1-deficiency, the reduced CD8+ T cell and neutrophil recruitment indicates that Tax1bp1-deficiency abrogates systemic inflammatory responses during infection.

**Tax1bp1-deficiency has a cell type-specific impact on *Mtb* growth.** Having shown that Tax1bp1-deficiency impacts the recruitment of immune cell subpopulations during *Mtb* infection, we then determined the effect of Tax1bp1-deficiency on *Mtb* growth in these same immune cells. As a reflection of *Mtb* burden, we analyzed the number of ZsGreen+ cells normalized to the number of each cell type analyzed. As expected, most of the green fluorescence signal from *Mtb* was detected in the myeloid cells (Fig 3B-D) that serve as a replicative niche for *Mtb*[4,9,48]. There were no significant differences in ZsGreen counts in T, B, or NK cells except at day 14 in CD8+ T cells (Fig 3B). Tax1bp1-deficiency reduced ZsGreen+ *Mtb* counts in AMs, MNC1, and MNC2 at 14 days post-infection and in AMs at 28 days post-infection (Fig. 3D). We interpret these results to suggest that Tax1bp1-deficiency restricts *Mtb* growth in AMs, MNC1, and MNC2 at specific time points.

To further investigate the hypothesis that Tax1bp1-deficiency restricts *Mtb* growth in a cell type-specific manner, we performed *Mtb* CFU analysis in sorted immune cells in addition to measurement of bacterial fluorescence as a proxy for *Mtb* burden. CFU analysis enables measurement of specifically live bacteria, whereas fluorescence counts can represent both live and dead bacteria. As opposed to the previous experiments analyzing immune cells from individual mice (Fig. 3), here we pooled lung cell suspensions from five mice of each genotype to obtain enough cells for downstream CFU analysis from sorted cell populations. In the rederived *Tax1bp1-/-* mice, as observed previously, Tax1bp1-deficiency restricted *Mtb* growth in the lungs and spleens (S8 Fig). Tax1bp1-deficiency also restricted *Mtb* CFU1bp1in innate immune cells except for neutrophils and MNC1 on day 21 (S8 Fig). To determine whether the cell type-specific phenotype of Tax1bp1-deficiency on *Mtb* growth was consistent across independent experiments, the ZsGreen+ *Mtb* counts from the sorted cells were quantified (experiment 1, S8 Fig) and analyzed together with a second independent experiment using pooled samples (experiment 2, S8 Fig), and the mean ZsGreen+ counts from mouse samples analyzed individually (experiment 3, S8 Fig) as previously displayed in Fig 3D. This analysis of three independent experiments indicates that Tax1bp1-deficiency can enhance *Mtb* growth in MNC1 and neutrophils at 21-days post-infection (S8 Fig), which is consistent with our previous CFU analysis in BMDMs infected *ex vivo*. In contrast, *Mtb* growth was reduced in *Tax1bp1-/-* AMs at 14- and 21-days post-infection and in MNC2 at 14-days post-infection (S8 Fig).

In summary, Tax1bp1-deficiency has a cell type-specific impact on *Mtb* growth, with an overall effect of decreasing *Mtb* growth in the major organs. We conclude that Tax1bp1-deficiency restricts *Mtb* growth in AMs and MNC2 while simultaneously enhancing *Mtb* growth in MNC1 and neutrophils at specific time points.

### *Ex vivo*

**Tax1bp1-deficiency reduces *Mtb* growth during AM infection *ex vivo*.** We were curious, given our previous *ex vivo* results in BMDMs [44] whether Tax1bp1-deficiency also abrogates *Mtb* growth in AMs infected *ex vivo*. Therefore,

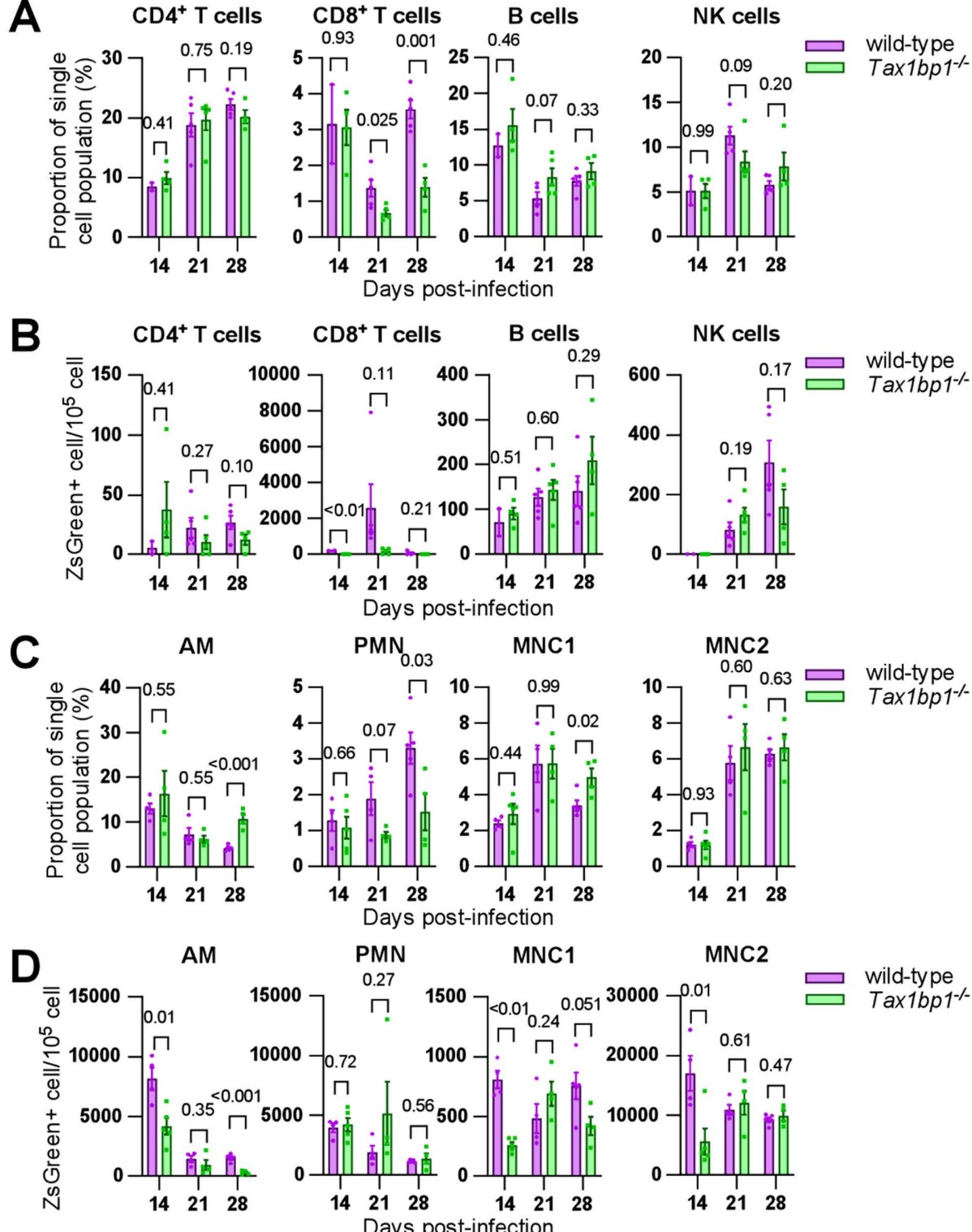

**Fig 3. Tax1bp1-deficiency impairs neutrophil and CD8+-T cell recruitment to the lungs during *Mtb* infection.** Lung homogenates from female mice (day 14, 21 post-infection) and male mice (day 28 post-infection) infected with ZsGreen-expressing *Mtb* were stained for antibodies to detect immune cells and analyzed by flow cytometry. **(A and C)** Displayed are the immune cell subtype proportions of the single cell population from individual

mouse lung samples. To account for differences in the number of cells analyzed between genotypes, data were normalized by dividing the proportion of each single cell population by the total number of each live cell type analyzed. **(B and D)** Displayed are the ZsGreen+ counts quantified in each immune cell subtype from individual mouse lung samples. Data were normalized to the number of cells sorted. Mean and SEM are displayed. The p-values from t-test comparisons are shown.

we obtained AMs from uninfected mice by bronchoalveolar lavage (BAL) that were isolated by simple adherence of the AMs in the BAL fluid. When infected with a luminescent strain of *Mtb, Mtb* grew more slowly in AMs from mice lacking Tax1bp1 relative to wild-type AMs both in the presence or absence of IFN-γ (Fig 4A),a cytokine that activates macrophage antibacterial host responses [56]. Similarly, luminescent and wild-type *Mtb* grew more slowly in *Tax1bp1^-/-* AMs as measured by CFU (Fig 4B-C).

Because adherent BAL cells may contain other cell types besides AMs that could influence microbial growth, we next characterized the cells in the BAL by flow cytometry and determined whether these growth phenotypes also occur in live CD11c⁺SilgecF⁺ AMs sorted from the BAL fluid. Consistent with a previous report that >95% of BAL cells from uninfected mice are AMs [57], 95.6% of live single cells from wild-type animals, and 96.8% of the cells from *Tax1bp1^-/-* mice, expressed the AM signatures CD11c and SiglecF (S9 Fig). Tax1bp1-deficiency reduced *Mtb* growth in the sorted AMs (S9 Fig). Together, this data implicates Tax1bp1-deficiency in reducing *Mtb* growth during AM infection and is consistent with our *in vivo* studies, although it is in sharp contrast to our results in *Mtb*-infected BMDMs *ex vivo* [44]. Therefore, we focused on understanding Tax1bp1's function in AMs as this model better represents the overall impact of Tax1bp1 on

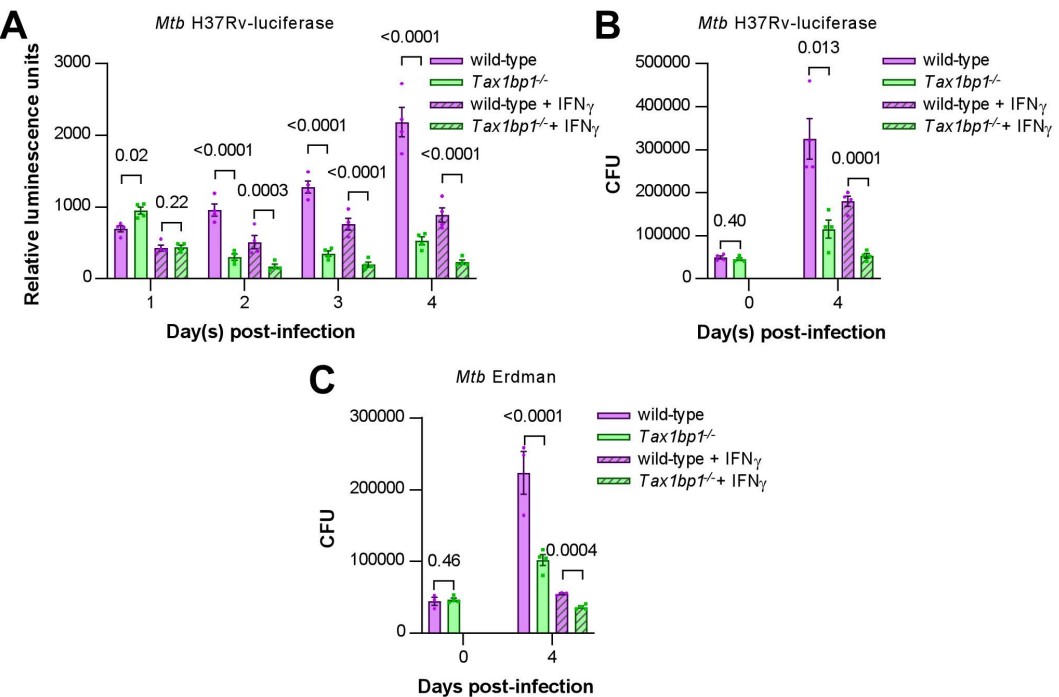

**Fig 4. *Mtb* growth was restricted in *Tax1bp1-/-* AMs infected *ex vivo*.** AMs were infected *ex vivo* with luciferase-expressing *Mtb* H37Rv **(A, B)** or wild-type *Mtb* Erdman **(C)** at a M.O.I. of 1 in the presence or absence of IFN-γ added at the time of infection. **(A)** The media was replaced with fresh media daily. Monolayer luminescence was measured daily. **(B, C)** CFU were measured immediately after infection (day 0) or 4 days post-infection. Displayed are the mean, SEM, and FDR-adjusted p values from the t-test.

real-world infections. Even though Tax1bp1's phenotype in BMDMs does not model the phenotype in several other cell types (*i.e.,* AMs, MNC2), we took advantage of Tax1bp1 phenotype in BMDMs by using BMDMs as a control.

**Tax1bp1-deficiency reduces autophagy flux and colocalization of LC3 with *Mtb*.** Tax1bp1 has an established role in antimicrobial selective autophagy [44, 45]. To determine whether this mechanism might contribute to differences in the growth of *Mtb* in *ex vivo* AMs and BMDMs from Tax1bp1-deficient mice relative to wild-type mice we investigated biogenesis of macrophage autophagosomes, which are double-membrane structures that form around cargo destined for lysosomal fusion. As an indicator of autophagosome formation, we measured the conversion of the precursor protein LC3-I to LC3-II [49]. LC3-II is the activated and cleaved form of LC3 that embeds in the autophagosome membrane [50]. Tax1bp1-deficiency led to a reduction in the LC3-II:-LC3-I ratio in unstimulated AMs (S10 Fig) but not BMDMs (S10 Fig), indicating that Tax1bp1-deficiency abrogates autophagosome formation in AMs. To assess autophagic turnover (flux), we increased the sensitivity of the assay by inducing LC3-I to LC-II conversion through starvation with nutrient-deficient media (EBSS) and blocked LC3-II degradation by treating the cells with bafilomycin (S10 Fig). Tax1bp1-deficiency led to a more dramatic decrease in the LC3-II:LC3-I ratio in AMs (S10 Fig) compared to BMDMs (S10 Fig), revealing that Tax1bp1-deficiency leads to a reduction in autophagy flux in AMs and BMDMs. These results are consistent with a previous report that Tax1bp1 promotes autophagy flux in HeLa cells [51].

Since we previously showed that Tax1bp1-deficiency impaired selective autophagy of *Mtb* in BMDMs [44] and Tax1bp1-deficiency affects autophagy flux in macrophages (S10 Fig)*,* we questioned whether Tax1bp1 would also reduce selective autophagy of *Mtb* in AMs. To test this, we performed confocal fluorescence microscopy of wild-type and *Tax1bp1-/-* AMs infected with fluorescent *Mtb* and assessed the colocalization of *Mtb* and autophagy markers*.* At 24 hours post-infection, Tax1bp1-deficiency slightly increased *Mtb* colocalization with ubiquitin by 16% (S11 Fig), whereas Tax1bp1-deficiency decreased *Mtb* colocalization with LC3 by 26% (S11 Fig). These results indicate that Tax1bp1-deficiency inhibits the progression of the ubiquitin+ *Mtb* phagosome to the LC3 + autophagosome. This is the same pattern of autophagy marker staining as we previously observed in *Mtb*-infected wild-type and *Tax1bp1-/-* BMDMs [44]. Since the impact of Tax1bp1-deficiency on *Mtb* growth, but not autophagy targeting, was cell type-specific, we reasoned that the lack of Tax1bp1 restricted *Mtb* growth in AMs by a different mechanism.

**Tax1bp1 protein level does not change during *Mtb* infection of AMs.** We next sought to identify a mechanistic explanation for the cell type-specific role of Tax1bp1-deficiency during *Mtb* infection of BMDMs and AMs. Based on previous literature that showed Tax1bp1 protein levels in peripheral blood mononuclear cells correlate with the presence of autoimmune disease [52], 8; and that Tax1bp1 protein expression is partially regulated at the post-translational level [58], we hypothesized that differences in Tax1bp1 protein abundance during *Mtb* infection may, in part, explain the different phenotypes in BMDMs and AMs.

Our previous global protein abundance measurements from cell lysate samples revealed that Tax1bp1 protein abundance increased by >2.5 fold in *Mtb*-infected wild-type BMDMs relative to mock-infected BMDMs after 24 hours [45]. Therefore, to compare with our previous analysis in BMDMs, we performed a similar proteomic analysis of five independent biological replicate wild-type AM cell lysate samples with the same *Mtb* strain and at the same multiplicity of infection (M.O.I) that were harvested at the same time point. Principal component analysis (PCA) revealed clustering of the data from the mock- or *Mtb*-infected replicate samples, indicating consistent changes in the data (S12 Fig). Using a $\log_2$(fold change *Mtb*-infected vs. mock-infected) threshold of >1 or <-1 and adjusted p value cut-off of 0.05, 144 proteins changed in protein abundance during *Mtb* infection of AMs (Fig 5A). As displayed in the Venn diagram, 70 of the proteins that changed in abundance in a statistically significant manner during *Mtb* infection of AMs were also identified during infection of BMDMs (Fig 5B). In contrast to our results in BMDMs, Tax1bp1 protein level did not significantly change during *Mtb* infection of wild-type AMs ($\log_2$(fold change) of 0.06, adjusted p value of 0.78; Fig 5A).

To determine the functional pathways that changed during *Mtb* infection, we performed gene ontology biological process enrichment analysis of the 70 proteins that changed in abundance significantly during infection in <u>both</u> AMs

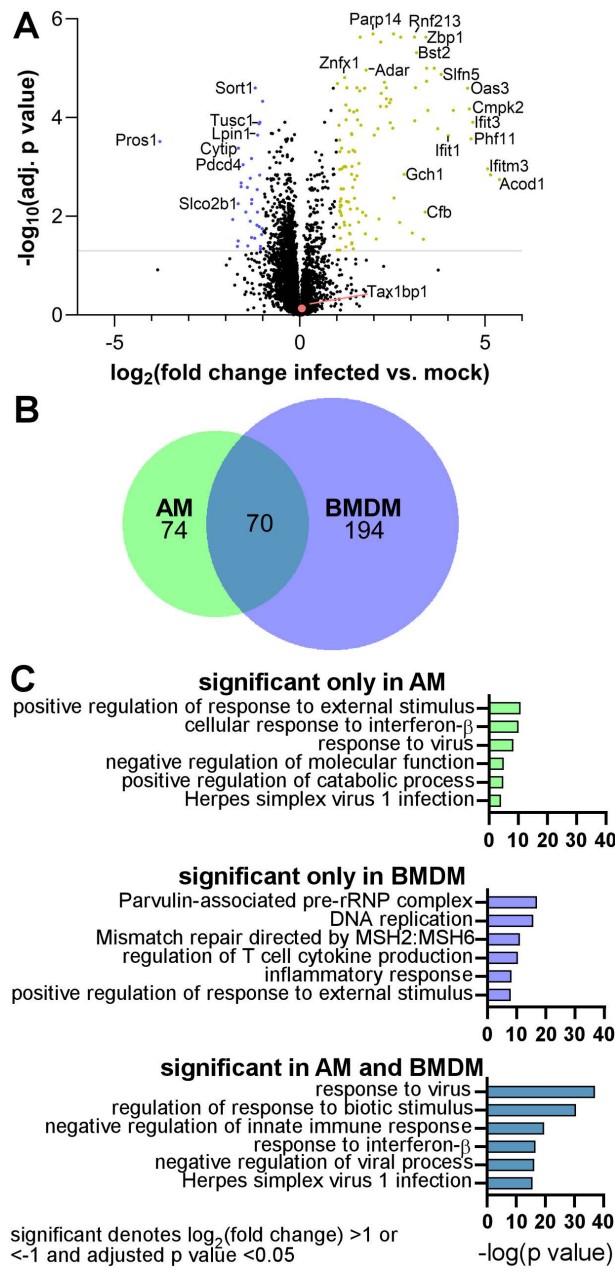

**Fig 5. Global protein abundance changes during *Mtb* infection of AMs.** In five independent biological replicate experiments, wild-type AMs were mock- or *Mtb*-infected at a M.O.I. of 10. At 24-hours post-infection, AM lysates were harvested, and the samples were digested with trypsin for LC-MS/MS analysis. The amino acid sequence, the relative levels of peptides, and the protein ID for the peptides were determined by a database search and statistical analysis in the infected vs. mock-infected samples. **(A)** Volcano plot displaying the proteins that significantly changed in abundance. The grey line denotes an adjusted p value of 0.05. Statistically significant hits with a $\log_2$fold change > 1 (yellow) or < -1 (blue) and an adjusted p value < 0.05 are highlighted. **(B)** Area-proportional Venn diagram displaying the number of statistically significant proteins that changed in abundance during *Mtb* infection of AMs and BMDMs. **(C)** Gene ontology enrichment analysis of statistically significant hits from global protein abundance analysis of the changes during *Mtb* infection of AMs, BMDMs, or both cell types. The enrichment analysis was performed with Metascape [59].

and BMDMs (Fig 5B). Then, to probe cell type-specific responses, we analyzed the 74 and 194 proteins that changed in abundance <u>only</u> in AMs and BMDMs, respectively (Fig 5B). The top six functional pathways identified in each group from the gene ontology enrichment analysis are displayed (Fig 5C). Consistent with *Mtb*'s ability to elicit anti-viral responses in macrophages [21], the response to virus pathway was the most significantly enriched pathway in both AMs and BMDMs (Fig 5B) and included proteins encoded by key interferon stimulated genes including *Isg15* [53], *Oasl2* [54] (analogous to human *Oasl* [55]), *Ifit1*, *Ifit2*, and *Ifit3* [60]. Pathways involved in DNA damage repair (mismatch repair directed by MSH2:MSH6), DNA replication, and ribosome biogenesis (parvulin-associated pre-rRNP complex) were significantly enriched in the analysis of proteins that changed in abundance during *Mtb* infection of BMDMs exclusively (Fig 5C). The proteins that changed in abundance only in AMs were encoded by genes enriched in viral and interferon response pathways, similar to those that significantly changed in both AMs and BMDMs (Fig 5C). In summary, although gene ontology enrichment analysis suggests that there are overlapping pathways in the host response to *Mtb* in both cell types, at the individual protein level there are differences in the responses between AMs and BMDMs. In particular, the increase in Tax1bp1 protein abundance during *Mtb* infection is cell-type specific, occurring in BMDMs but not AMs. The proteomic analysis highlights mechanistic differences in the host responses to *Mtb* in these two cell types.

**Host and pathogen expression analysis of *Mtb*-infected *Tax1bp1-/-* AMs *ex vivo*.** We exploited the observation that *Mtb* infection of AMs *ex vivo* recapitulates the phenotype seen *in vivo* in Tax1bp1-deficient mice by using an unbiased transcriptomics approach to develop hypotheses about host responses regulated by Tax1bp1. To broadly query host effector responses in Tax1bp1-deficient cells and simultaneously determine if the lack of Tax1bp1 triggers downregulation of *Mtb* genes that support intracellular *Mtb* replication, we performed host and pathogen dual transcriptional profiling of *Mtb*-infected wild-type and *Tax1bp1-/-* AMs. Differential gene expression analysis of *Mtb* transcripts showed no significant changes in gene expression using an adjusted p value cutoff of 0.05. However, using a less stringent p value threshold for statistical significance of an unadjusted p value of 0.05, similar to other reports of *Mtb* transcriptional profiling [61], Tax1bp1-deficiency downregulated two *Mtb* genes, *mmpL4* and *mbtE*, following *Mtb* infection (S13 Fig). While the former is required for intracellular *Mtb* replication [62], we conclude that Tax1bp1-deficiency does not lead to major changes in the *Mtb* transcriptional profile at this time point of 36 hours post-infection. One possible explanation for the discrepancy between changes in *Mtb* growth caused by Tax1bp1-deficiency and the lack of major *Mtb* transcriptional changes is that our transcriptional profiling was performed before significant changes in *Mtb* growth were observed, beginning at 48 hours post-infection (Fig 4). Another possibility is that Tax1bp1-deficiency led to changes in host responses that increase the killing of *Mtb* by host cells because this would not be expected to produce dramatic changes in the bacterial transcriptional profile.

Consistent with the idea that Tax1bp1-deficiency significantly alters host responses to *Mtb*, we identified 140 differentially expressed host genes with a $log_2$(fold change) of greater than 1 or less than -1 and an adjusted p value <0.05 comparing wild-type and *Tax1bp1-/-* AMs infected with *Mtb* (S13 Fig). Gene ontogeny enrichment analysis of the up- and down-regulated genes identified cytokines and inflammatory response (20 genes, -log (p value) of 4.8) as a significant functional pathway controlled by Tax1bp1 in *Mtb*-infected AMs. Genes in the cytokines and inflammatory response pathway included genes downregulated (*e.g., Cxcl1* [63]) and upregulated (*e.g., Cd4* [64], *Pf4* [65], *Pdgfa* [56], *Kitl* [57], and *Cxcl3* [66]) in *Tax1bp1-/-* AMs (Figs 6 and S13). Gene ontogeny enrichment analysis also identified positive regulation of the intrinsic apoptotic signaling pathway (16 genes, -log (p value) of 4.1), including the genes encoding for Bok, an apoptotic regulator, and prostaglandin-endoperoxide synthase 2 (*Ptgs2*, also known as COX-2; S13 Fig). Most of the genes in this pathway, including *Mmp2* [67], *Nck2* [67], *Bok* [68], *Gsdme* [69], *Cam2kb* [58], *Eya4* [70], *Pmp22* [71], and *Rgcc* [72], were upregulated in *Tax1bp1-/-* AMs. However, *Ptgs2* [73], *Clu* [61], *Ctgf* [62], and *Cxcl1* [63] were four other genes in this pathway that were negatively regulated in *Tax1bp1-/-* AMs. The differentially expressed gene of the greatest magnitude upregulated in *Mtb*-infected *Tax1bp1-/-* AMs compared to wild-type AMs was *Sox7* (S13 Fig; $log_2$(fold change) of 5.4, adjusted p value of $2.94 \times 10^{-12}$). Sox7 is a transcription factor that induces apoptosis through the MAP kinase ERK-BIM

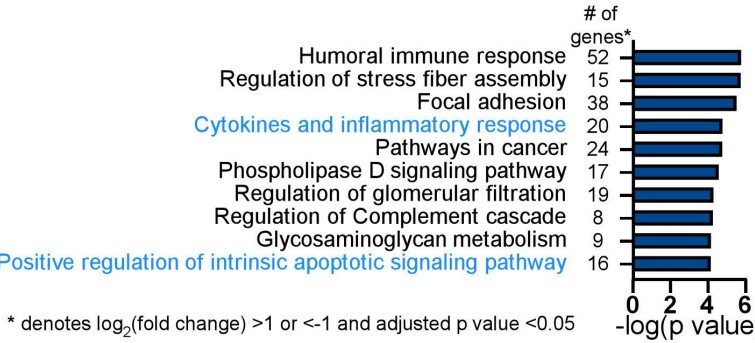

**Fig 6. Tax1bp1-deficiency results in differential expression of inflammatory response and apoptotic signaling pathway genes during *Mtb* infection of AMs.** Wild-type and *Tax1bp1⁻/⁻* AMs were infected in biological triplicate with *Mtb* at a M.O.I. of 2. The RNA was harvested at 36-hours post-infection for differential pathogen and host gene expression analysis by RNAseq. Gene ontogeny enrichment analysis of statistically significant differentially expressed host genes (log$_2$(fold change) >1 or <-1, adj. p values <0.05) during *Mtb* infection of wild-type and *Tax1bp1⁻/⁻* AMs was performed with Metascape [59]. Among this analysis of the up- and down-regulated genes triggered by Tax1bp1-deficiency, the top ten enriched pathways and the number of genes in each functional pathway are displayed.

(BCL2-interacting mediator of cell death) pathway [74]. In summary, gene ontogeny enrichment analysis suggests that Tax1bp1-deficiency regulates the expression of inflammatory signaling and host cell death genes, both of which can contribute to the control of *Mtb* growth. Therefore, we further tested the hypothesis that Tax1bp1-deficiency impacts these two host responses during *Mtb* infection of AMs.

**Tax1bp1-deficiency reduces inflammatory cytokine signaling during AM infection *ex vivo*.** Given that inflammatory responses are intricate and the impact of particular genes or proteins can vary based on the cell type and immune environment [75], we next investigated whether the effects of Tax1bp1-deficiency on cytokines and inflammatory responses during AM infection *ex vivo* aligned with our observations *in vivo* (Figs 1C, 1D, and S1). In our RNAseq analysis, there were trends towards increased gene expression of IL-1β(2.3 fold change in wild-type vs. *Tax1bp1⁻/⁻* AMs, adjusted p value of 0.073) and interleukin-1 receptor type 1 (*Il1r1*; 2.1 fold change in wild-type vs. *Tax1bp1⁻/⁻* AMs, adjusted p value of 0.053; S13 Fig) in wild-type AMs.

To determine if Tax1bp1-deficiency impacts cytokine responses at the protein level and to elicit more robust cytokine responses, we next performed *Mtb* infections at a higher M.O.I. of 5. Tax1bp1-deficiency led to decreased levels of several inflammatory cytokines, including IL-1β, but not IFN-β, in infected cell supernatants (Fig 7A). As previously noted,

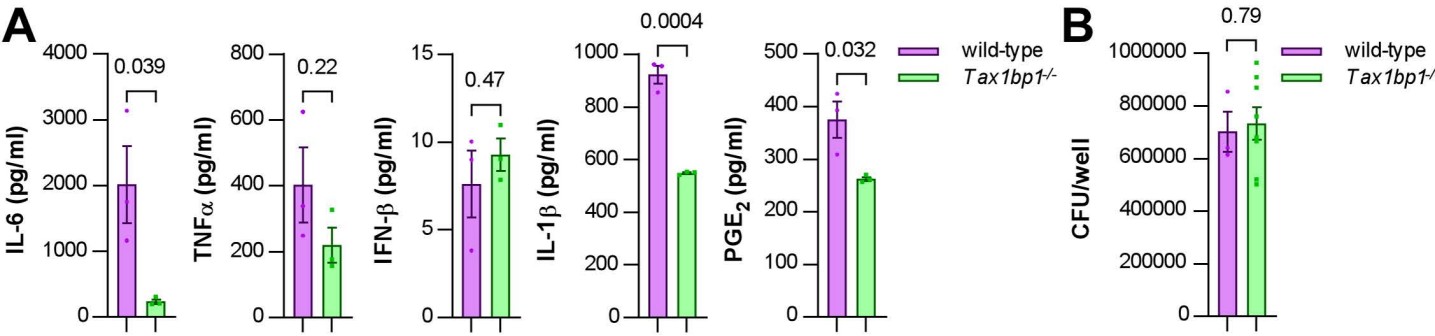

**Fig 7. IL-6, IL-1β, and PGE$_2$ production was abrogated during AM infection of *Tax1bp1-/-* AMs.** AMs were seeded at 100,000 cells/well and infected in triplicate wells with *Mtb* at a **M.**O.I. of 5. At 24 hours post-infection, **(A)** the supernatants were collected for cytokine measurement by ELISA, and **(B)** AM monolayers were lysed and plated for *Mtb* CFU. Mean, SEM, and p values from the t-test are displayed.

Tax1bp1-deficiency also led to decreased expression of *Ptgs2* (prostaglandin-endoperoxide synthase 2; 4.2 fold change in wild-type vs. *Tax1bp1⁻/⁻* AMs, adjusted p value of 0.01; S13 Fig), which is involved in the production of inflammatory prostaglandins. Notably, prostaglandin $E_2$ ($PGE_2$) is an inflammatory eicosanoid that decreases *Mtb* killing by blocking efferocytosis [76,77]. In addition to reducing levels of cytokines regulated by NF-κB, Tax1bp1-deficiency also reduced the production of $PGE_2$ during *Mtb* infection of AMs (Fig 7A). To our knowledge, $PGE_2$ was not previously known to be regulated by Tax1bp1. Importantly, these cytokine and eicosanoid levels were measured early after infection (24 hours) when the *Mtb* CFU were the same in wild-type and *Tax1bp1⁻/⁻* AMs (Fig 7B), before any differences in *Mtb* growth were observed at later time points. Therefore, these findings suggest that the increased cytokine and $PGE_2$ production mediated by Tax1bp1-deficiency happens independently of bacterial burden. In summary, Tax1bp1-deficiency decreased the production of proinflammatory cytokines and eicosanoids during *Mtb* infection of AMs. These findings are consistent with our cytokine analysis during *in vivo* infection (Figs 1 and 2) and in contrast to our previous report in BMDMs, in which Tax1bp1-deficiency did not impact inflammatory cytokine production during *Mtb* infection [45].

**Tax1bp1 enhances necrotic-like cell death and delays apoptosis of *Mtb*- infected AMs.** In addition to regulating cytokine and inflammatory responses, our gene expression analysis suggested that Tax1bp1-deficiency regulates apoptotic gene expression during AM infection. Tax1bp1 was previously shown to negatively regulate apoptosis following cytokine stimulation [78] and viral infection [43]. Furthermore, apoptosis is a mode of cell death that leads to killing of *Mtb* through efferocytosis, in which *Mtb*-infected apoptotic cells are phagocytosed by neighboring macrophages [79, 80]. This is in contrast to necrosis, which is a form of uncontrolled cell death that is highly immunostimulatory and enhances *Mtb* growth [81–85].

To test if Tax1bp1-deficiency impacts cell death during *Mtb* infection of AMs and BMDMs, we analyzed uninfected control cells and *Mtb*-infected cells by live cell fluorescence microscopy using CellEvent caspase 3/7 to detect apoptotic cells and propidium iodide (PI) for necrotic/late apoptotic cells. In the absence of IFN-γ stimulation, Tax1bp1-deficiency led to a modest decrease in necrotic-like cell death of AMs on days 1–4 post-infection (Figs 8A, 8B and S14). In IFN-γ stimulated AMs, Tax1bp1-deficiency accelerated apoptosis on day 3 post-infection (Figs 8C, 8D and S15). In contrast to AMs, Tax1bp1-deficiency did not impact the amount of apoptotic or necrotic-like cell death in unstimulated BMDMs (S15A-S15C, S16A and S16B Figs). Only when stimulated with IFN-γ, Tax1bp1-deficiency accelerated apoptotic cell death during *Mtb* infection of BMDMs on day 4 post-infection (S15 Fig). Together, these results show that Tax1bp1-deficiency reduces necrotic-like cell death of *Mtb*-infected AMs (Fig 8E) but not BMDMs, and Tax1bp1-deficiency advances the timing of apoptosis during *Mtb* infection of IFN-γ-stimulated AMs and BMDMs. Only the latter phenotype in IFN-γ-stimulated *Tax1bp1⁻/⁻* AMs was not specific to *Mtb*-infection because we also observed enhanced apoptosis in the uninfected IFN-γ-stimulated *Tax1bp1⁻/⁻* AMs (S16 Fig).

## Discussion

The autophagy receptor, Tax1bp1, plays a role in multiple stages of intracellular pathogen infections, including autophagy and regulation of cytokine responses. More specifically, Tax1bp1 was implicated in the termination of inflammatory NF-κB signaling during Sendai virus and VSV infection [40] as well as the restriction of *Mtb* growth in BMDMs [44]. Here we show that Tax1bp1-deficiency, unexpectedly, also plays a role in limiting inflammation *in vivo* and in reducing *Mtb* growth in AMs and MNC2 (Fig 3). While Tax1bp1 did not impact *Listeria* growth in macrophage *ex vivo*, Tax1bp1-deficiency hinders *Listeria* and *Mtb* infection in mice (Figs 1 and 2). In addition to *Mtb* and *Listeria*, differing results between *ex vivo* and *in vivo* pathogen replication were also reported during RSV infection [38]. Viral replication is restricted during *Tax1bp1⁻/⁻* murine infection but only slightly changed in cultured A549 *Tax1bp1* knockdown cells [38]. Collectively, the differences in pathogen replication in mice and cultured cells suggest that the cell type and the tissue environment *in vivo* can play a critical role in the function of Tax1bp1. Indeed, transplanted bone marrow precursors and terminally differentiated macrophages can change their chromatin landscape in various tissue environments [86]. This has significant consequences in the lungs,

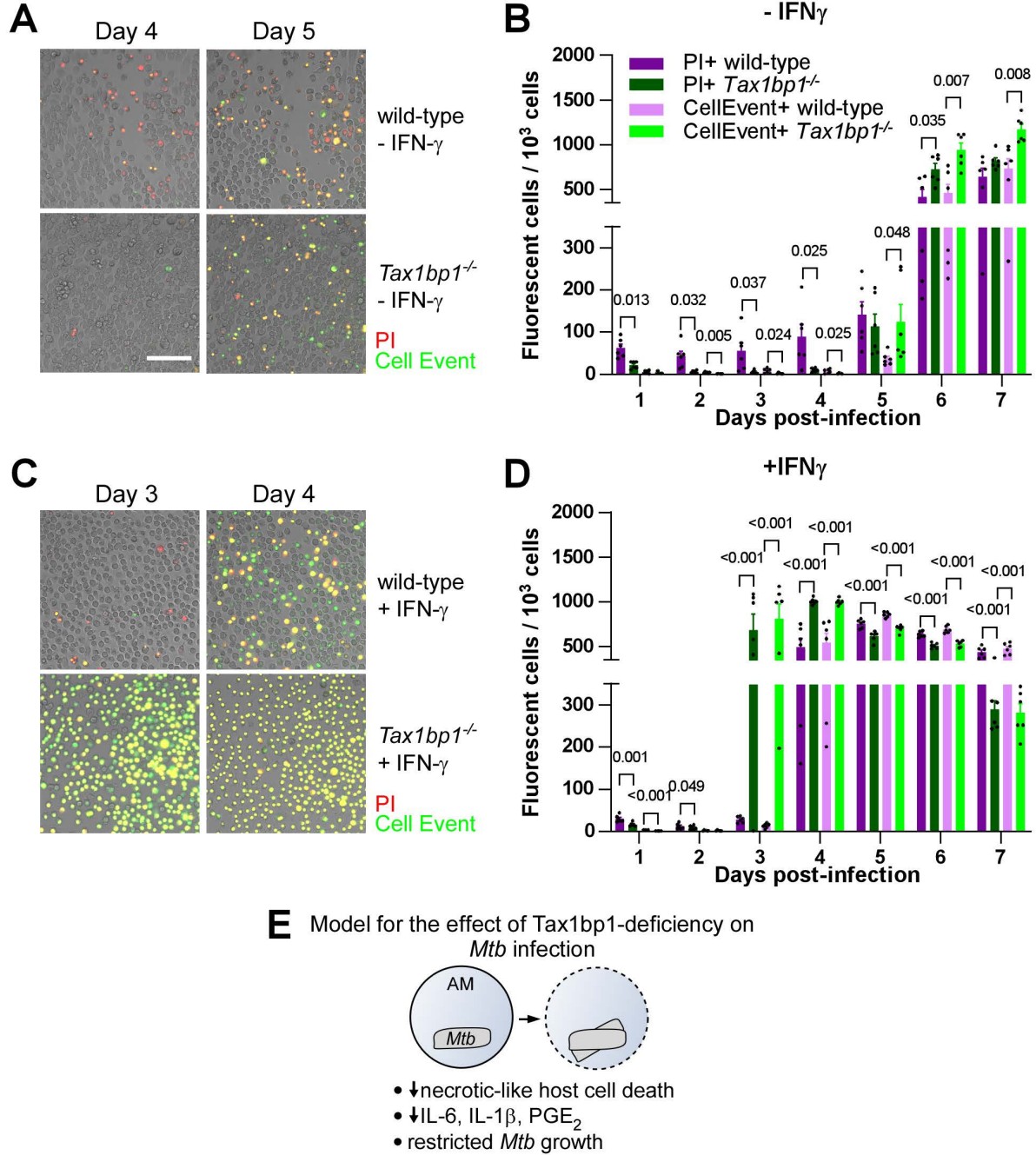

**Fig 8. Necrotic-like cell death was inhibited and apoptosis accelerated in *Mtb*-infected *Tax1bp1-/-* AMs.** AMs were infected with *Mtb* at a M.O.I. of 1 in the presence of CellEvent caspase 3/7 to detect apoptotic cells and propidium iodide (PI) for necrotic/late apoptotic cells without **(A, B)** or with **(C, D)** IFN-γ added to the media. Fluorescence images were obtained at 20X magnification in two positions per well in three replicate wells. **(A, C)** Representative fluorescence and brightfield microscopy images were merged, cropped, and scaled. **(B, D)** The number of fluorescent cells in each field was quantified in the green (CellEvent) and red fluorescence (PI) channels. Fluorescent cell numbers were normalized by the total number of cells in the phase contrast image for each field. Mean, SEM, and statistically significant FDR-adjusted p values from t-test comparisons are displayed. For clarity, only statistically significant p values (p < 0.05) are shown. The white bar depicts 100 μm. Data are representative of three independent experiments. **(E)** Tax1bp1-deficiency restricts *Mtb* growth, inflammatory cytokine synthesis, PGE$_2$ production, and necrotic-like host cell death in AMs.

where the microenvironment impacts macrophage activation and function [87], and impacts key host responses to *Mtb*. For example, during *Mtb* infection of macrophages *ex vivo*, macrophages up-regulate pro-inflammatory genes mediated by type I IFN and NFκB [21, 27, 88, 89]; however, during *Mtb* infection *in vivo,* AMs display a relatively anti-inflammatory phenotype [6]. The discovery that Tax1bp1-deficiency reduces *Mtb* growth in AMs and MNC2 implies that Tax1bp1-deficiency restricts *Mtb* replication in several innate immune cell types. In contrast, Tax1bp1-deficiency promoted *Mtb* replication in BMDMs, MNC1, and neutrophils at specific time points.

We discovered that one difference between BMDMs and AMs is that Tax1bp1 protein abundance increased during *Mtb* infection of the former but not the latter. Since Tax1bp1 expression levels play a major role in its ability to clear protein aggregates induced by stressors [46], this insight may in part explain the different phenotypes mediated by Tax1bp1 during *Mtb* infection of BMDMs and AMs. Another difference between BMDMs and AMs is that M-CSF [macrophage colony stimulating factor [90, 91] is used as a stimulating factor for the differentiation of the former, whereas GM-CSF [75,92,93] is thought to be more crucial for the differentiation and maintenance of the latter [76, 77]. Since these stimulating factors induce phenotypic changes in macrophages [94] and GM-CSF can be a bactericidal effector against *Mtb* [95], testing whether the stimulating factor present during immune cell differentiation enables Tax1bp1 to promote or restrict *Mtb* growth may shed light on an underlying mechanism that drives Tax1bp1's cell type-specific function.

In addition to inducing inflammatory signaling during infection *ex vivo*, we discovered that Tax1bp1-deficiency initially reduces necrotic-like cell death in the first four days of *Mtb* infection in AMs but not BMDMs. In the presence of IFN-γ, Tax1bp1-deficiency accelerates apoptosis during *Mtb* infection of AMs. Since *Mtb* growth is enhanced in necrotic macrophages [85, 96] and apoptosis leads to restriction of *Mtb* growth via efferocytosis [80]*,* these results indicate that the impact of Tax1bp1-deficiency on the mode of host cell death is a mechanism by which Tax1bp1 enhances *Mtb* growth in AMs. Indeed, Tax1bp1 was previously shown to mediate ferroptosis, a programmed type of necrosis, in response to copper stress-induced reactive oxygen species [97]. Interestingly, ferroptosis enhances *Mtb* dissemination [98]. In addition to promoting ferroptosis, Tax1bp1 is known to impact apoptotic signaling by restraining apoptosis during VSV and Sendai virus infection [43]. During viral infection, termination of RIG-I mediated mitochondrial antiviral signaling proteins (MAVS) signaling blocked apoptosis and type I IFN signaling [43]. Because we did not observe altered levels of type I IFN during AM infection with *Mtb*, we hypothesize Tax1bp1 signals through a different pathway during *Mtb* infection in these cells. Tax1bp1 also blocks apoptosis by acting as an adaptor for TNFAIP3 (also known as A20) to bind and inactivate its substrates RIPK1 (receptor (TNFRSF)-interacting serine-threonine kinase 1) in the TNFR signaling pathway [99]. Further experiments are needed to determine the downstream signaling pathway by which Tax1bp1-deficiency induces apoptosis and whether the lack of Tax1bp1 inhibits a programmed form of necrosis during *Mtb* infection.

We acknowledge that a limitation of our mechanistic studies in AMs infected *ex vivo* is that we have not determined whether these same cell death phenotypes also occur in AMs during animal *Mtb* infection or in 100% pure populations of AMs sorted from the BAL fluid. Thus, it is possible that the cell death and inflammatory responses may be influenced by other contaminating cells in the BAL fluid. Recent work has highlighted that monocyte-derived AMs (moAMs), a CD11c-SiglecF-pro-inflammatory macrophage that expresses other alveolar macrophage markers, are recruited to the lung during the course of *Mtb* infection [100, 101]. However, these cells are absent from naïve (uninfected) mice [100], which we used as a source of AMs for *Mtb* infections *ex vivo*. Thus, effects from moAMs are not a possible confounder in our *ex vivo* infection experiments.

Tax1bp1 is an autophagy adaptor, and autophagy plays a role in cell-autonomous immunity to microbial pathogens. Autophagy also affects immune cell development and inflammatory responses [102–106]. The *Tax1bp1*-deficient knockout mice used in this study are deficient in *Tax1bp1* in all cells. While our results indicate that the effect of Tax1bp1-deficiency is mediated during the innate and adaptive immune responses, other cells may likely require Tax1bp1 for their function, which might impact *Mtb* infection. For example, Tax1bp1 is important for the metabolic transition of activated T cells [107]. Tax1bp1 also terminates ERK signaling in B cells to mediate B cell differentiation and antigen-specific antibody production

[108]. Therefore, in addition to being involved in the recruitment of CD8+ T cells and neutrophils to the lung during *Mtb* infection, Tax1bp1 may be needed for the normal function of other immune cells that undergo proliferation and activation in addition to AMs. Indeed, the possible effects of Tax1bp1-deficiency on other cell types may play an important role in contributing to host mortality during later stages of infection and disease.

Another potential limitation of this study is our use of a Tax1bp1 whole-body deficient mouse line generated by the replacement of the last exon of Tax1bp1 with a neomycin cassette [37]. These mice still express a truncated Tax1bp1 allele lacking the C-terminal zinc finger domains [46]. We believe the use of these mice is valid because these domains enable the protein's ubiquitin-dependent functions [46], although the truncated Tax1bp1 protein could hypothetically retain some partial activity. However, resolving this and the effects of Tax1bp1 on specific cell types would require genetic complementation in Tax1bp1 mutant cells or a mouse strain in which temporal or cell-specific deletion of Tax1bp1 would be possible.

In conclusion, Tax1bp1-deficiency plays a unique role in controlling host cell death and limiting inflammatory responses during *Mtb* and *Listeria* infection, in contrast to the observation that Tax1bp1-deficiency enhances NF-κB signaling during viral infection. We discovered that Tax1bp1 is a host factor contributing to differences in *Mtb* growth in AMs compared to BMDMs [6]. While multiple autophagy receptors, including Tax1bp1 and p62, target *Mtb* for selective autophagy [23, 44], this work reveals that different autophagy receptors may play fundamentally distinct roles in *Mtb* pathogenesis, even depending on the host cell type. In contrast to Tax1bp1, the primary autophagy receptor, p62, is not involved in survival from *Mtb* infection [109]. These findings would suggest that one could alter the inflammatory responses and augment protective host responses to pathogens by blocking Tax1bp1 function. A better understanding of the mechanism by which Tax1bp1 regulates host cell death, autophagy, and inflammation during infection may enable the development of Tax1bp1 as a target for anti-bacterial therapies.

## Methods

### Ethics statement

Animal infections were performed in accordance with the animal use protocol (AUP-2015-11-8096, AN192778–01) approved by the Animal Care and Use Committee at the University of California, Berkeley, and the Institutional Animal Care and Use Program at the University of California, San Francisco, in adherence with the federal regulations provided by the National Research Council and National Institutes of Health.

### *M. tuberculosis* mouse infections at UC Berkeley

*Tax1bp1-/-* mice were provided by Dr. Hidekatsu Iha, Oita University, Japan. Low-dose aerosol infection (100 CFU) of age- and sex-matched wild-type or Tax1bp1-deficient mice (male and female, age 8–12 weeks) was performed with *M. tuberculosis* Erdman strain using the Glass-Col Inhalation Exposure System. One day after infection, infected mice were euthanized, the lungs were homogenized, and CFU were enumerated on 7H10 agar plates supplemented with 10% OADC and 0.5% glycerol to determine the inoculum. On days 9, 11, 21, and 50 after infection, the lung was divided into portions. The superior lobe of the right lung was fixed in 10% buffered formalin for histologic analysis. The remainder of the right lung and left lung were combined and homogenized in 1 ml of PBS containing 0.05% Tween80 in a Bullet Blender Tissue Homogenizer (Next Advance). The spleen and liver were homogenized in 400 µl or 2 ml of PBS containing 0.05% Tween80, respectively. For measurement of CFU, organ homogenates were serially diluted and plated on 7H10 agar plates supplemented with 10% OADC and 0.5% glycerol. For measurement of cytokines from the lung homogenate, protein extraction was performed by combining 700 µl of homogenate with 700 µl of Tissue Protein Extraction Reagent (T-PER; Thermo Scientific) containing cOmplete, mini, EDTA-free protease inhibitor cocktail (Roche 11836170001) at 2X concentration. Samples were incubated for 20 min at 4°C, vortexed, and centrifuged at 12,000 x g for 10 min. The

supernatants were filter sterilized with a 0.22 μm filter and stored at -80°C until further analysis. For survival experiments, mice were sacrificed after 15% loss of maximum body weight.

**Cytokine measurements.** Cytokines from lung lysates were measured using DuoSet ELISA kits (R&D systems) following the manufacturer's protocol. Interferon levels were measured from the *Mtb*-infected lung lysates using L929 ISRE-luciferase reporter cells as previously described [110]. Luciferase reporter cells were seeded in a 96-well plate for 24 hours. Lung lysates were incubated with the reporter cells for 8 hours, and luciferase activity was measured with the Luciferase Assay Report Assay (Promega) using the manufacturer's protocol.

**Histology sample processing and quantitative analysis.** Formalin-fixed specimens were washed three times in PBS and stored in 70% ethanol. Histologic processing was performed by Histowiz. Serial ultrathin sections were stained for hematoxylin & eosin (H&E), ubiquitin (anti-ubiquitylated antibody, AB1690, EMD-Millipore, 1:100 dilution), tuberculosis (Abcam ab214721, 1:1000 dilution), or myeloperoxidase (Abcam ab9535, 1:50 dilution). Primary antibodies were detected with 3,3'-diaminobenzidine (DAB) staining. The ubiquitin and tuberculosis IHC images were aligned and combined using image registration scripts in QuPath. The MPO staining analysis was analyzed using Indica Labs Halo image analysis software. Cells were segmented using the Multiplex IHC algorithm v3.1.4 in Halo and MPO-positive cells were determined by thresholding the DAB channel. Positive cells were further sub-divided into Low intensity, Medium Intensity and High intensity bins to allow for subsequent calculation of a H-score based on percentage of cells positive for Low, Medium and High using the following formula: H-Score = (1 x % positive cells low) + (2 x % positive cells medium) + (3 x % positive cells high). Cells in all the intensity bins were considered positive for myeloperoxidase staining.

Images of tuberculosis lesions were exported from QuPath into ImageJ. A minimum threshold of two standard deviations above the mean signal was applied to filter positive pixels for ubiquitin and tuberculosis immunohistochemistry images. Colocalization was calculated from pixel overlap in images of the tuberculosis and ubiquitin immunohistochemistry. A veterinary pathologist analyzed the H&E histopathology images.

***M. tuberculosis* mouse infections at UC San Francisco.** *Tax1bp1*$^{-/-}$ mice were imported from UC Berkeley and rederived by the UC San Francisco Rederivation Core to eliminate the potential for any interinstitutional murine pathogen transmission. Low-dose aerosol infections (100–200 CFU) of age- and sex-matched wild-type and *Tax1bp1*$^{-/-}$ mice were performed with a Glass-col inhalation chamber. At the indicated time points, mice were euthanized with $CO_2$, and their lungs were minced with scissors and digested in 3 ml of RPMI-1640 with 5% heat-inactivated FBS containing 1 mg/ml collagenase D (Sigma) and 30 μg/ml DNAseI (Sigma) for 30 min at 37°C. Cells were processed with a gentleMACS dissociator (Miltenyi Biotec, lung program 2) and filtered through a 70 μm strainer. The samples were rinsed with 1 ml of FACS buffer (PBS with 3% heat-inactivated FBS, 2 mM EDTA). Residual tissue on the cell strainer was further processed using a syringe plunger and rinsed with 1 ml of FACS buffer. The cell suspension was then centrifuged at 650 × g for 3 minutes at 4°C, and the supernatant was discarded. The cell pellet was resuspended in 3 ml of ACK lysis buffer (Gibco) to lyse the RBCs, and lysis was quenched with 3 ml of FACS buffer solution. After centrifuging the cell suspension at 650 × g for 3 minutes at 4°C, the supernatant was removed, and the cells were resuspended in 1 ml of FACS buffer. Each cell suspension was pooled (from 5 mice) and passed through a 50-μm strainer.

Single-cell lung suspensions were stained with the Zombie Aqua Fixable Viability kit (1:200 dilution, BioLegend, #423101) and treated with CD16/CD32 Fc block (1:100 dilution, BD 553142) in PBS (1 ml) for 15 min. The samples were centrifuged at 650 × g for 3 minutes at 4°C, and the supernatants were removed. Cells were stained with 2 ml of antibody mixture diluted in Brilliant Stain Buffer (Table 1; Invitrogen #00-4409-42) for 30 min at 4°C. Antibodies diluted in Brilliant Stain Buffer (BD, #566349) were added to the cells. Antibody staining was performed for 30 minutes at 4°C. Subsequently, the cell suspensions were centrifuged at 650 × g for 3 minutes at 4°C, washed with 1 ml of FACS buffer solution, resuspended in 3 ml of FACS buffer, and passed through a 50-μm strainer. Cell subsets were sorted using a BD Aria Fusion Sorter through a 100 μm nozzle using the 4-way purity mode.

**Table 1. Key Resources.**

| Reagent type (species) or resource | Designation | Source or reference | Identifiers | Additional information |
|---|---|---|---|---|
| Biological sample (*M. musculus*) | Primary bone marrow-derived macrophages, AMs, and peritoneal exudate cells, wild-type C57BL/6J | Jackson Laboratory | Stock # 000664 | |
| Biological sample (*M. musculus*) | Primary bone marrow-derived macrophages, AMs, and peritoneal exudate cells, *Tax1bp1*-/- | [37] | *Tax1bp1*-/- | |
| Antibody | Anti-ubiquitylated antibody | EMD-Millipore | AB1690 | 1:100 dilution |
| Antibody | Anti-*Mycobacterium tuberculosis* antibody | Abcam | ab214721 | 1:1000 dilution |
| Antibody | Anti-Myeloperoxidase antibody | Abcam | ab9535 | 1:50 dilution |
| Antibody | Anti-LC3 (2G6) | NanoTools | 0260-100/LC3-2G6 | 1:200 |
| Antibody | Anti-ubiquitinylated proteins antibody (FK2) | Sigma-Aldrich | 04-263 | 1:400 |
| Antibody | PE Rat Anti-Mouse Siglec-F | BD Biosciences | BDB552126 | 1:200 |
| Antibody | Anti-mouse CD16/CD32 Fc block | BD Biosciences | BD553142 | 1:100 |
| Antibody | PE/Cyanine5 anti-mouse CD90.2 (Thy1.2) | BioLegend | 105314 | 1:300 |
| Antibody | PE/Cyanine5 anti-mouse CD19 | BioLegend | 115510 | 1:300 |
| Antibody | PE/Cyanine5 anti-mouse NK-1.1 | BioLegend | 108716 | 1:200 |
| Antibody | PE/Cyanine7 anti-mouse Ly-6C | BioLegend | 128018 | 1:300 |
| Antibody | Brilliant Violet 421 anti-mouse Ly-6G | BioLegend | 127628 | 1:200 |
| Antibody | Brilliant Violet 605 anti-mouse CD11c | BioLegend | 117334 | 1:200 |
| Antibody | Brilliant Violet 711 anti-mouse/human CD11b | BioLegend | 101242 | 1:200 |
| Antibody | Alexa Fluor 647 anti-mouse MHCII | BioLegend | 107618 | 1:300 |
| Antibody | PE/Cyanine7 anti-mouse CD4 | Biolegend | 100528 | 1:400 |
| Antibody | Brilliant Violent 605 anti-mouse CD8a | Biolegend | 100744 | 1:200 |
| Antibody | Alexa Fluor 647 anti-mouse NK1.1 | Biolegend | 108720 | 1:200 |
| Antibody | APC-Fire750 anti-mouse CD3 | Biolegend | 100248 | 1:200 |
| Commercial assay | Luciferase assay system | Promega | E1500 | |
| Commercial kit | Mouse IL-6 DuoSet ELISA | R and D | DY406 | |
| Commercial kit | Mouse TNF-α DuoSet ELISA | R and D | DY410 | |
| Commercial kit | Mouse IL-12/IL-23 p40 allele-specific DuoSet ELISA | R and D | DY499 | |
| Commercial kit | Cytometric Bead Array Mouse Inflammation Kit | BD Biosciences | 552364 | |
| Commercial kit | Mouse IFN-β ELISA Kit, high sensitivity | PBL Assay Science | 42410-1 | |
| Commercial kit | Prostaglandin $E_2$ Express ELISA Kit | Cayman Chemicals | 500141 | |
| Commercial kit | Q5 Site Directed Mutagenesis kit | New England Biolabs | E0554S | |
| Commercial kit | NEBuilder HiFi DNA Assembly Cloning Kit | New England Biolabs | E5520S | |
| Commercial kit | LR Clonase II Enzyme Mix | Invitrogen | 11791020 | |
| Dye | Zombie Aqua | BioLegend | 423102 | 1:200 |
| Strain | Lenti-X 293T | TakaraBio | 632180 | |
| Strain | *Mtb*: Erdman | ATCC | 35801 | |
| Strain | *Listeria monocytogenes* 10403S | [111] | | |
| Strain | *Mtb*: H37Rv (pMV306hsp-LuxG13) | Plasmid from Addgene | 26161 | |
| Strain | *Mtb*; Erdman (pMV261:: ZsGreen) | [9] | | |

Bacteria were quantified from sorted cells by serial dilution in PBS containing 0.05% Tween80 and plated on 7H10 agar plates supplemented with 10% Middlebrook OADC, 0.5% glycerol, and PANTA antibiotic mixture at a 1:500 dilution to reduce contamination risk from non-mycobacteria during organ dissection. BD PANTA antibiotic mixture (BD, B4345114) containing polymyxin B, amphotericin B, nalidixic acid, trimethoprim, and azlocillin was prepared by dissolving the contents of 1 lyophilized vial in 3 ml of OADC.

**Bone marrow-derived macrophage infection.** Bone marrow-derived macrophage infections with *Listeria monocytogenes* 10403S were performed as previously described [112–114].

**Peritoneal cell exudate infection.** Approximately 7 ml of ice-cold PBS was injected into the peritoneum of euthanized mice. Peritoneal exudate cells were treated with ACK lysis buffer, resuspended in tissue culture cell media (RPMI supplemented with 1 mM L-glutamine and 10% fetal bovine serum), and seeded into 24-well plates with glass coverslips at a density of $1.5 \times 10^6$ cells/ml for 24-hours prior to infection. Prior to infection, non-adherent cells were removed by replacement of the tissue culture media.

*Listeria monocytogenes* 10403S was inoculated from a single colony into BHI media. Following overnight incubation at 30 °C, bacteria were washed in PBS and resuspended to an OD of 1.5 in PBS. Bacteria were diluted 1:1000 in tissue culture cell media for infection of peritoneal exudate cells. 30 minutes post-infection, cells were rinsed twice with PBS, and fresh media was replaced. At 1 hour post-infection, gentamicin sulfate was added at a final concentration of 50 µg/ml to kill extracellular bacteria. At 2- and 8 hours post-infection, coverslips were placed in 5 ml of water and vortexed. Serial dilutions were plated on LB agar supplemented with streptomycin (200 µg/ml). CFU were enumerated after 24 hours of incubation at 37 °C. Infections were performed with 3 coverslips for each experimental condition.

**_L. monocytogenes_ mouse infections.** Age and sex-matched male and female mice were infected with *L. monocytogenes*. A 2 ml overnight culture of *L. monocytogenes* grown in brain heart infusion media at 30 °C slanted. *L. monocytogenes* was subcultured in 5.5 ml of BHI media and incubated with shaking at 37 °C until reaching an optical density between 0.4-0.8. The bacteria were washed and diluted in PBS to achieve an inoculum of approximately $5 \times 10^5$ CFU/ml. 200 µl of this suspension was injected into the tail vein of the mice. For the intraperitoneal infection, the bacterial cells were prepared as described, and 200 µl of bacterial suspension was injected into the peritoneum at a dose of $3.74 \times 10^5$ CFU/mouse. At the indicated time points, the spleen and liver were harvested in water containing 0.1% NP-40. Organs were homogenized, and CFU were enumerated on LB agar supplemented with streptomycin (200 µg/ml).

**Alveolar macrophage (AM) isolation and culture.** AMs were harvested from naïve mice by bronchoalveolar lavage with 10 ml of PBS containing 2 mM EDTA, 0.5% fetal bovine serum (FBS) pre-warmed to 37°C as described previously [115, 116]. BAL cells were seeded at a density of 100,000 cells/well in 96-well plates. For experiments with sorted AMs, live, CD11c+SiglecF+AMs were sorted at a density of 50,000 cells/well in 96-well plates using the previously described protocol for antibody staining and gating [117]. For short-term cultivation up to 4 days, AMs were cultured in RPMI-1640 medium supplemented with 10% (v/v) FBS, 2 mM GlutaMAX, 10 mM HEPES and 100 U ml–1 penicillin–streptomycin. After allowing at least 2 hours for adhesion, the media was replaced with fresh media without antibiotics. For cultivation >4 days, AMs were cultured in RPMI-1640 medium supplemented with 10% (v/v) FBS, 2 mM GlutaMAX, 1 mM sodium pyruvate, and 2% (v/v) GM-CSF supernatant produced by a B16 murine melanoma cell line.

**Macrophage infections with _Mtb_.** *Mtb* H37Rv strain was transformed with pMV306hsp-LuxG13 for expression of *LuxCDABE*. Logarithmic phase cultures of *Mtb* (H37Rv-Lux or wild-type Erdman strain) were grown in 7H9 media supplemented with 10% Middlebrook OADC, 0.5% glycerol, 0.05% Tween80 in inkwell bottles at 37°C with rotation at 100 rpm. *Mtb* cell pellets were washed twice with PBS followed by centrifugation for 5 min at $1462 \times g$ and sonication to remove and disperse clumps. *Mtb* was resuspended in RPMI with 10% horse serum. Media was removed from the macrophage monolayers, the bacterial suspension was overlaid, and centrifugation was performed for 10 min at $162 \times g$. Following infection, the media was replaced with cultivation media with 15 ng/ml IFN-γ (Peprotech) or without IFN-γ. In experiments performed with luminescent *Mtb*, luminescence measurements were obtained daily following media changes

daily using a GloMax microplate reader (Promega). For CFU measurements, the monolayers were washed with PBS, lysed in PBS with 0.05% Tween80, serially diluted, and spread on 7H10 agar plates supplemented with 10% Middlebrook OADC and 0.5% glycerol. CFU were enumerated after 21 days of incubation at 37°C.

**Gene expression analysis during *Mtb* infection of AMs.** AMs were infected with wild-type *Mtb* Erdman at a M.O.I. of 2. At 36-hours post-infection, the monolayers were washed with PBS, and the AMs were lysed in 200 μl of Trizol reagent. Samples were pooled from four technical replicate wells. The experiment was performed independently three times (*i.e.,* three independent biological replicates). *Mtb* was centrifuged, the supernatant containing host RNA was removed, and the *Mtb* pellet was resuspended in 400 μl of fresh Trizol and 0.1 mm zirconia/silica beads. *Mtb* was mechanically disrupted with the Mini Bead-Beater Plus (Biospec Products) as previously described [118]. 70% of the sample containing host RNA was pooled with the sample containing *Mtb* RNA, the samples were treated with 200 μl of chloroform, and RNA was purified with the Trizol Plus RNA purification kit (Ambion). Purified total RNA was treated with DNAseI (ThermoFisher) and dried by rotary evaporation in RNA stabilization tubes (Azenta US, Inc.; South Plainfield, NJ, USA). Sample QC, dual rRNA depletion for bacteria and mouse, library preparation, Illumina sequencing (2x150 bp; 30M reads per sample), and differential gene expression analysis were performed by Azenta Life Sciences US, Inc.

### Sample QC

Total RNA samples were quantified using Qubit 2.0 Fluorometer (Life Technologies, Carlsbad, CA, USA) and RNA integrity was checked with 4200 TapeStation (Agilent Technologies, Palo Alto, CA, USA).

### Library Preparation and Sequencing

ERCC RNA Spike-In Mix kit (cat. 4456740) from ThermoFisher Scientific was added to normalized total RNA prior to library preparation following manufacturer's protocol. rRNA depletion was performed using QIAGEN FastSelect rRNA Bacteria + HMR Kit or HMR/Bacteria (Qiagen, Germantown, MD, USA), which was conducted following the manufacturer's protocol. RNA sequencing libraries were constructed with the NEBNext Ultra II RNA Library Preparation Kit for Illumina by following the manufacturer's recommendations. Briefly, enriched RNAs are fragmented for 15 minutes at 94°C. First strand and second strand cDNA are subsequently synthesized. cDNA fragments are end repaired and adenylated at 3'ends, and universal adapters are ligated to cDNA fragments, followed by index addition and library enrichment with limited cycle PCR. Sequencing libraries were validated using the Agilent Tapestation 4200 (Agilent Technologies, Palo Alto, CA, USA), and quantified using Qubit 2.0 Fluorometer (ThermoFisher Scientific, Waltham, MA, USA) as well as by quantitative PCR (KAPA Biosystems, Wilmington, MA, USA).

The sequencing libraries were multiplexed and clustered onto a flowcell on the Illumina NovaSeq instrument according to manufacturer's instructions. The samples were sequenced using a 2x150bp Paired End (PE) configuration. Image analysis and base calling were conducted by the NovaSeq Control Software (NCS). Raw sequence data (.bcl files) generated from Illumina NovaSeq was converted into fastq files and de-multiplexed using Illumina bcl2fastq 2.20 software. One mismatch was allowed for index sequence identification.

### Data Analysis

After investigating the quality of the raw data, sequence reads were trimmed to remove possible adapter sequences and nucleotides with poor quality using Trimmomatic v.0.36. The trimmed reads were mapped to the *Mus musculus* and *Mtb* Erdman strain reference genomes available on ENSEMBL using the STAR aligner v.2.5.2b. BAM files were generated as a result of this step. Unique gene hit counts were calculated by using feature Counts from the Subread package v.1.5.2. Only unique reads that fell within exon regions were counted.

Using DESeq2, a comparison of gene expression between the groups of samples was performed. The Wald test was used to generate p values and Log2 fold changes. Genes with adjusted p values $< 0.05$ and absolute $\log_2$ fold changes $>1$ were called as differentially expressed genes for each comparison.

**Cytokine analysis of *Mtb*-infected AMs.** AMs were infected with wild-type *Mtb* Erdman strain at a M.O.I. of 5. At 24 hours post-infection, the supernatants were filtered through a 0.2 μm syringe filter and analyzed by ELISA for IFN-β (PBL Assay Bioscience), TNF-α, and IL-1β (R&D systems) as previously described, and prostaglandin $E_2$ (Cayman Chemicals).

**Live cell imaging.** Uninfected control macrophages and macrophages infected with *Mtb* at a M.O.I. of 1 were incubated in the presence of 0.1 μg ml$^{-1}$ of propidium iodide (LifeTechnologies) and two drops per milliliter of CellEvent Caspase-3/7Green ReadyProbes reagent (Invitrogen) to measure necrosis/late apoptosis and apoptosis, respectively. After the initial media change on the day of *Mtb* infection, as previously performed [85], the cell culture media was not changed during the remainder of the experiment to avoid disruption of the macrophage monolayers. Fluorescence and phase contrast images were obtained at 20x magnification with a Keyence BZ-X 700 microscope. Images were obtained daily in three technical replicate wells per condition and at two positions in each well. Quantification of the number of necrotic and apoptotic cells was performed with ImageJ version 1.54f as described previously [85]. Images were converted to 8-bit (grayscale), binarized, and enumerated using the analyze particles module (size threshold 0.001-infinity). Cell numbers from phase contrast microscopy images were enumerated using Cell Profiler.

**Immunofluorescence microscopy.** AMs were infected with fluorescent *Mtb* at a M.O.I. of 2. At 8- and 24-hours post-infection, monolayers were washed with PBS, fixed with 4% PFA for 20 minutes, washed with PBS, and stained with anti-LC3 or anti-ubiquitin primary antibodies and AlexaFluor-647 conjugated secondary antibodies as previously described [44]. Images were obtained at 63x magnification from quadruplicate wells per condition, in 69 x/y positions, and 4 z positions (0 μm, 0.5 μm, 1 μm, and 1.5 μm) with an Opera Phenix microscope (Perkin Elmer). Colocalization analysis of LC3, ubiquitin, and *Mtb* was performed with Harmony version 4.9 (Perkin Elmer) using the following analysis parameters. The four z stack images in each x/y position were processed into a maximum intensity projection. Nuclei were identified in the DAPI channel using Method B with a common threshold of 0.07 and an area threshold of $> 20$ μm$^2$. Cytoplasm was identified in the AlexaFluor 647 channel using Method A with an individual threshold of 0.06. The find spot module was used to identify LC3 or ubiquitin "spots" in the AlexaFluor 647 channel using method C with a contrast setting of 0.42, uncorrected spot to region intensity of 3.8, and default radius. *Mtb* were identified in the AlexaFluor 488 channel using the find spot module method B with a detection sensitivity of 0.5 and splitting sensitivity of 0.5. To identify *Mtb* that colocalized with LC3 or ubiquitin "spots", the select population module was used for the *Mtb* population with the select by mask method. The percent colocalization was calculated for each well from all the images obtained in each well using the evaluation module.

**LC3 immunoblot.** AMs and BMDMs were either incubated in fresh culture media or Earle's Balanced Salt Solution (EBSS) for 8 hours following a previously described protocol [85]. For cells treated with EBSS, 300 nM bafilomycin was added for 2 hours prior to the completion of treatment. Cell lysates were generated by adding RIPA buffer containing mini-complete protease inhibitor cocktail (Roche). Lysate protein concentration was determined by BCA assay. 15 μg of protein from each sample was separated by SDS-polyacrylamide gel electrophoresis (NuPage 4–12% Bis-Tris Mini-protein gels) and transferred to PVDF membranes. After blocking with Intercept (PBS) blocking buffer (LicorBio #927–70001), membranes were probed with anti-LC3 antibodies (Sigma L7543, dilution 1:1000) followed by IRDye 800CW Goat anti-rabbit IgG secondary antibody (LicorBio #926–32211, dilution 1:10,000). The membrane was then probed anti-pan actin antibodies (Cell Signaling #4968, dilution 1:1000) followed by IRDye 800CW Goat anti-rabbit IgG secondary antibody. Membrane images were obtained on a Li-Cor Odyssey system. Band densities for LC3-I and LC3-II were measured using ImageJ (National Institutes of Health).

**AM infections for global protein abundance analysis.** BAL was performed on 14 mice per biological replicate experiment. BAL cells were pooled and seeded at a density of 40e$^4$-60e$^4$ cells/well in 4-well Omniwell plates in AM media

containing GMCSF. Media was changed daily. After 7 days of incubation at 37°C with 5% $CO_2$, half of the AMs were infected with *Mtb* Erdman strain at a M.O.I. of 10. The other half of the AMs were mock-infected by changing the media with RPMI-1640 with 10% heat-inactivated horse serum and centrifuged simultaneously with the *Mtb*-infected AMs. After centrifugation, the media was changed to AM media. After 24-hours incubation at 37°C with 5% $CO_2$, the monolayers were washed with PBS, and AMs fixed on the plates with 100% methanol for 20 minutes. The monolayers were washed twice with PBS. AMs were lysed with 6 ml of lysis buffer (8 M urea, 150 mM NaCl, 100 mM ammonium bicarbonate, pH 8; added per 10 ml of buffer: 1 tablet of Roche mini-complete protease inhibitor EDTA free and 1 tablet of Roche PhosSTOP tablet) prepared fresh before each replicate experiment. Lysates were stored at −80°C until further processing.

**Trypsin digest and desalting.** A bicinchoninic acid assay (Pierce) was performed to measure protein concentration in cell lysate supernatants. 1 mg of each clarified lysate was reduced by the addition of dithiothreitol (DTT) to a 5 mM final concentration for 30 min at room temperature and alkylated by the addition of iodoacetamide to 10 mM final concentration for 30 min at room temperature in the dark. Remaining alkylating agent was quenched by the addition of DTT to 10 mM final concentration for 30 min at room temperature in the dark. The samples were diluted with 100 mM ammonium bicarbonate, pH 8.0, to reduce the urea concentration to below 2M. Samples were incubated with sequencing grade modified trypsin (Promega) at a 1:25 enzyme:protein ratio and LysC (Wako) at a 1:100 enzyme:protein ratio overnight at 37°C with rotation. Prior to desalting, the sample pH was reduced to approximately 2.0 by the addition of 10% trifluoroacetic acid (TFA) to a final concentration of 1% trifluoroacetic acid. Peptides were desalted using 50 mg SepPak C18 solid-phase extraction cartridges (Waters). The columns were activated with 3 ml of 80% acetonitrile (ACN) 0.1% TFA, and equilibrated with 3 ml of 0.1% TFA. Peptide samples were applied to the columns, and the columns were washed with 3 ml of 0.1% TFA. Peptides were eluted with 1.1 ml of 40% ACN, 0.1% TFA. 10 µg of peptides analyzed by LC-MS/MS for global protein abundance measurement.

**Liquid chromatography and mass spectrometry.** Following digestion, peptides were separated on a PepSep reverse-phase C18 column (1.9 µm particles, 15 cm, 150 mm ID) (Bruker) with a gradient of 3–28% buffer B (0.1% formic acid in 80% acetonitrile) over buffer A (0.1% formic acid in water) over 67 minutes, an increase to 40% B in 5 minutes, and held at 95% B for 8 minutes. Eluting peptide cations were analyzed by electrospray ionization on an Orbitrap Exploris (Thermo Fisher Scientific). For DIA analysis, MS1 scans of peptide precursors were performed at 120,000 resolution (200 m/z) over a scan range of 350–1050 m/z, with an AGC target of 250% and a max injection time of 100 ms. MS2 scans were collected over 350–950 m/z in 15 m/z isolation windows with a 0.5 m/z overlap. Maximum injection time was set to auto, with AGC set to standard, and an MS2 resolution of 15,000. Higher energy collisional dissociation (HCD) was performed at a NCE of 28%. Six gas-phase fractions (GPF)-DIA runs were collected from a pooled sample of all conditions [119]. tSIM MS1 scans were performed at 120,000. tMS2 scans were collected at 30,000 fragment resolution, an AGC target of 1e6, a maximum ion injection time of 60 ms, and a NCE of 26. Data was collected using 4 *m/z* precursor isolation windows in a staggered-window pattern, with each GPF fraction covering an approximately 100 m/z range.

**Global protein abundance analysis.** The GPF mass spectrometry data was searched against the Uniprot mouse database appended with the Erdman *Mtb* strain in Spectronaut to build a spectral library of detected peptides and proteins (Biognosys, version 19.0). Default settings, including trypsin digestion, variable modifications of methionine oxidation and N-termini acetylation, and fixed modification of cysteine carbamidomethylation, were used. DIA mass spectrometry data was then searched in Spectronaut against the GPF-generated spectral library. DIA runs were filtered to obtain a false discovery rate of 1% at the peptide spectrum match and protein level [120]. Quantitative analysis was performed in the R statistical programming language using the artMS package (release 3.20) and quantitative packages from MSstats (version 3.20) [121]. For MSstats analysis, the normalization method was set to equalizeMedians, and censored missing values were imputed by the Accelerated Failure Time mode. Statistically significant changes were selected by applying a log2-fold-change (>1.0 or<−1.0) and an adjusted p value (<0.05) [122].

**Statistical analysis.** GraphPad Prism (v.10.3.1) was used for statistical analysis. Unpaired t-test comparisons were calculated assuming Gaussian distributions, and the p values were reported. In experiments with more than two experimental conditions, p values from the t-test comparison between two groups were adjusted for the FDR (multiple comparisons) using the two-stage linear step-up procedure of Benjamini, Krieger, and Yekutieli (Figs 4, 8, S15, and S16).

## Supporting Information

**S1 Text. Supporting information figure legends.**
(DOCX)

**S1 Fig. Tax1bp1-deficiency abrogates *M. tuberculosis* virulence and inflammatory cytokine responses.**
(TIF)

**S2 Fig. Analysis of lung pathology and neutrophil staining in the lungs during *M. tuberculosis* aerosol infection of wild-type and *Tax1bp1*-/- mice.**
(TIF)

**S3 Fig. Ubiquitin colocalization with *M. tuberculosis* in the lungs during murine aerosol infection of wild-type and *Tax1bp1*-/- mice.**
(TIF)

**S4 Fig. Tax1bp1-deficiency limits *L. monocytogenes* growth during murine infection.**
(TIF)

**S5 Fig. Tax1bp1-deficiency decreases the frequency of microabscess formation and lymphocyte depletion during *L. monocytogenes* infection.**
(TIF)

**S6 Fig. Gating strategy used for the identification of myeloid subsets.**
(TIF)

**S7 Fig. Gating strategy used for the identification of NK, T, and B cells.**
(TIF)

**S8 Fig. Tax1bp1-deficiency reduced *Mtb* growth in AMs and MNC2.**
(TIF)

**S9 Fig. Tax1bp1-deficiency restricts *Mtb* growth in sorted AMs.**
(TIF)

**S10 Fig. Tax1bp1-deficiency reduces autophagy flux.**
(TIF)

**S11 Fig. Tax1bp1-deficiency abrogates the colocalization of *Mtb* with LC3.**
(TIF)

**S12 Fig. Principal component analysis (PCA) of changes in global protein abundance during *Mtb*- or mock-infection of AMs.**
(TIF)

**S13 Fig. Pathogen and host differential gene expression analysis volcano plots.**
(TIF)

**S14 Fig. Tax1bp1-deficiency reduces necrotic-like cell death and accelerates apoptosis during *Mtb* infection of AMs.**
(TIF)

**S15 Fig. Tax1bp1-deficiency accelerates apoptosis during *Mtb* infection of IFN-γ-stimulated BMDMs.**
(TIF)

**S16 Fig. Tax1bp1-deficiency induces apoptosis in uninfected IFN-γ-stimulated AMs.**
(TIF)

## Acknowledgments

We acknowledge Dr. Gabe Murphy, Research Development Strategist and Grant Writer from the UCSF Office of Collaborative Research, for assistance in manuscript editing. We acknowledge members of Professor Shaeri Mukherjee's lab, Ady Steinbach and Thomas Moss, for assistance in creating Cell Profiler pipelines for the enumeration of cell numbers in phase contrast microscopy images. We acknowledge support from Professor Daniel A. Portnoy for infections with *Listeria monocytogenes* and the Biological Imaging Development CoLab at UCSF for assistance in developing a pipeline in QuPath for ubiquitin colocalization analysis with *Mtb* in immunohistochemistry images. Experimental data was produced in The PCAT at UCSF.

## Author contributions

**Conceptualization:** Jeffrey Chin, Alicia Richards, Danielle L Swaney, Nevan J Krogan, Joel D. Ernst, Jeffery S Cox, Jonathan M. Budzik.

**Data curation:** Jeffrey Chin, Nalin Abeydeera, Vinh Q Nguyen, Jonathan M. Budzik.

**Formal analysis:** Jeffrey Chin, Nalin Abeydeera, Teresa Repasy, Rafael Rivera-Lugo, Gabriel Mitchell, Vinh Q Nguyen, Jeffery S Cox, Jonathan M. Budzik.

**Funding acquisition:** Nevan J Krogan, Joel D. Ernst, Jeffery S Cox, Jonathan M. Budzik.

**Investigation:** Jeffrey Chin, Nalin Abeydeera, Teresa Repasy, Rafael Rivera-Lugo, Gabriel Mitchell, Vinh Q Nguyen, Weihao Zheng, Alicia Richards, Erica Stevenson, Danielle L Swaney, Jonathan M. Budzik.

**Methodology:** Jeffrey Chin, Nalin Abeydeera, Teresa Repasy, Rafael Rivera-Lugo, Gabriel Mitchell, Vinh Q Nguyen, Weihao Zheng, Joel D. Ernst, Jeffery S Cox, Jonathan M. Budzik.

**Project administration:** Jeffery S Cox, Jonathan M. Budzik.

**Resources:** Vinh Q Nguyen, Joel D. Ernst, Jeffery S Cox, Jonathan M. Budzik.

**Supervision:** Jeffery S Cox, Jonathan M. Budzik.

**Validation:** Jonathan M. Budzik.

**Visualization:** Jeffery S Cox, Jonathan M. Budzik.

**Writing – original draft:** Jeffery S Cox, Jonathan M. Budzik.

**Writing – review & editing:** Jeffrey Chin, Nalin Abeydeera, Rafael Rivera-Lugo, Gabriel Mitchell, Vinh Q Nguyen, Weihao Zheng, Alicia Richards, Danielle L Swaney, Nevan J Krogan, Joel D. Ernst, Jeffery S Cox, Jonathan M. Budzik.

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
