## [Decision Letter · Decision Letter 0]

26 Feb 2025

Tax1bp1 enhances bacterial virulence and promotes inflammatory responses during Mycobacterium tuberculosis infection of alveolar macrophages

PLOS Pathogens

Dear Dr. Budzik,

Thank you for submitting your manuscript to PLOS Pathogens. After careful consideration, we feel that it has merit but does not fully meet PLOS Pathogens's publication criteria as it currently stands. Therefore, we invite you to submit a revised version of the manuscript that addresses the points raised during the review process.

Please submit your revised manuscript within 60 days Apr 27 2025 11:59PM. If you will need more time than this to complete your revisions, please reply to this message or contact the journal office at plospathogens@plos.org. Please include the following items when submitting your revised manuscript:

We look forward to receiving your revised manuscript.

Kind regards,

Pierre Santucci

Academic Editor

PLOS Pathogens

Helena Boshoff

Section Editor

PLOS Pathogens

Editor-in-Chief

PLOS Pathogens

orcid.org/0000-0003-2946-9497

Michael Malim

Editor-in-Chief

PLOS Pathogens

orcid.org/0000-0002-7699-2064

**Additional Editor Comments :**

Dear Dr. Budzik,

Thank you for submitting your manuscript PPATHOGENS-D-24-02731 entitled "Tax1bp1 enhances bacterial virulence and promotes inflammatory responses during Mycobacterium tuberculosis infection of alveolar macrophages" to PLOS Pathogens. Your work has now been assessed by three independent experts and we have completed the review process.

As you will see, the three experts find the topic and the approach of your research to be of significant interest and a potential great contribution to the field, however there are several aspects that require substantial revision before the manuscript can be considered for publication.

As we feel that the number of requested experiments might be too important, we have outlined some key concerns and suggestions provided by the reviewers that need to be addressed in your revised manuscript. In addition, we expect you to carefully refine your conclusions according to the reviewers' reports and address each of their comments in a point-by-point response prior resubmission.

Summary of the Reviewers' Evaluation

The three independent experts think that the paper is overall interesting, however they report a lack of mechanistical insights regarding the authors previous observations, reporting that TAX1BP1 restricts Mtb replication in BMDM and the current manuscript that supports a model in which TAX1BP1 act as pro-Mtb factor in vivo. They feel that the discrepancies have not been adequately addressed/biologically explained. Therefore, addressing this part appears to be important, several key experiments have been suggested below.

In addition, the three referees raised also concerns about consistency, standardisation and performing the appropriate number of replicates per experiments. The three experts report that MOI are changing in between experiments, in vivo timepoints are not always the same across experiments, and some experiments have not been done in triplicate. Please address them accordingly.

Reviewers suggested that the in vivo phenotype – that differs from the one observed in BMDM in their previous publication - might be caused by systemic inflammatory regulation rather than localized macrophage dysfunction and/or tissue damage, and that such process might involve alternative cell types. Accordingly, such hypothesis could be addressed by performing more thorough characterisation of the cell populations contained within the lesions at different timepoints in WT and TAX1BP1 KO infected mice. Please carefully inspect what they suggested, and perform additional experiments accordingly. Priority should be given to complementing the H&E histopathological analysis, with flow cytometry-based quantification of the cell populations within the lesions by focusing on CD4 and CD8 T cells but also neutrophils.

Reviewers had concerned about collecting cells via BAL, which has numerous limitations. Therefore, they suggested to perform a thorough characterisation of the cells contained within the BAL as they can be highly heterogeneous. They also suggested to go further, and purify cells according to their Siglec profiles and subsequently perform the experiments to test your hypothesis on purified tissue-resident alveolar macrophages only.

Finally, they suggested to perform more in-depth characterisation of TAX1BP1 role in autophagosome formation, endolysosomal fusion and phagosomal rupture/cytosolic access. Experimental strategies based on Electron or fluorescence microscopy have been suggested to fully characterise the potential underlying role of TAX1BP1 in Mtb intracellular lifestyle and replication. However, after a consultation with other editors we feel that focusing mainly on the in vivo work might be more appropriate to strenghten the paper. Therefore, we feel that adressing these comments with experimental work is optional.

Of note, one of the reviewer raised concerns about the restricted access to the RNA-seq dataset as the raw data seem to be inaccessible until 2028. Authors should provide access to their data at least using a specific restricted access for reviewers only.

**Journal Requirements:**

2) We noticed that you used the phrase 'data not shown' in the manuscript. We do not allow these references, as the PLOS data access policy requires that all data be either published with the manuscript or made available in a publicly accessible database. Please amend the supplementary material to include the referenced data or remove the references.

4) Please check the Supplementary Figure files legends within the manuscript. Currently, "Figure 8-figure supplement 1" is duplicated in the Supplementary Figures legends while " Figure 9-figure supplement 1" is missing. Please ensure that the Supplementary Figures are labeled in a consecutive numerical order as For example, “S1 Figure” and “S2 Figure,”  and so forth.

5) We notice that your supplementary Figures are included in the manuscript file. Please remove them and upload them with the file type 'Supporting Information'. Please ensure that each Supporting Information file has a legend listed in the manuscript after the references list.

Potential Copyright Issues:

i) Figure 9-figure supplement 1. Please confirm whether you drew the images / clip-art within the figure panels by hand. If you did not draw the images, please provide (a) a link to the source of the images or icons and their license / terms of use; or (b) written permission from the copyright holder to publish the images or icons under our CC BY 4.0 license. Alternatively, you may replace the images with open source alternatives. See these open source resources you may use to replace images / clip-art:

7) Thank you for stating that "Confocal microscopy images are available on the Dryad repository (DOI: 10.5061/dryad.44j0zpcq6). " Should your submission be accepted, we will require the following information in your Data Availability Statement:

1. The DOI provided by Dryad

2. The citation for your data package in the reference section of your manuscript

3. The citation for your data package in the methods section

If you are unable to adhere to our open data policy, please kindly revise your statement to explain your reasoning and we will seek the editor's input on an exemption. Please be assured that, once you have provided your new statement, the assessment of your exemption will not hold up the peer review process.

8) In the online submission form, you indicated that "Flow cytometry data files are available upon request." All PLOS journals now require all data underlying the findings described in their manuscript to be freely available to other researchers, either

1. In a public repository

2. Within the manuscript itself

3. Uploaded as supplementary information.

9) Please amend your detailed Financial Disclosure statement. This is published with the article. It must therefore be completed in full sentences and contain the exact wording you wish to be published.

10) Please ensure that the funders and grant numbers match between the Financial Disclosure field and the Funding Information tab in your submission form. Note that the funders must be provided in the same order in both places as well.

Currently, these funds "Mentored Scientist in Tuberculosis Award (R25AI147375), TB RAMP program (R25AI147375), and the University of California Dissertation-Year Fellowship" are missing from the Funding Information tab.

**Reviewers' Comments:**

Reviewer's Responses to Questions

**Part I - Summary**

Reviewer #1: This article investigates the critical role of the autophagy receptor Tax1bp1 in inflammatory responses and host susceptibility to infections by Mycobacterium tuberculosis (Mtb) and Listeria monocytogenes. The findings reveal a dual functional role of Tax1bp1: it restricts bacterial growth in bone marrow-derived macrophages (BMDMs) but promotes bacterial replication in alveolar macrophages (AMs), neutrophils, and specific monocyte-derived subsets from the bone marrow.

Tax1bp1 influences necrotic-like cell death in AMs, facilitating Mtb replication, in contrast to its restrictive effect in BMDMs. Furthermore, either Tax1bp1 deficiency or expression of a phosphosite-deficient version of Tax1bp1 restricts Mtb growth. The study demonstrates that Tax1bp1 links autophagy, cell death regulation, and pro-inflammatory responses, with cell type-specific effects on bacterial growth control.

This article presents intriguing and relevant findings on the role of Tax1bp1 in innate immune responses and bacterial infection progression. However, to strengthen its scientific impact, several points should be clarified, and additional experiments are recommended.

Reviewer #2: This manuscript investigates the role of Tax1bp1 in the regulation of Mycobacterium tuberculosis (Mtb) infection, with a focus on its contributions to inflammatory cytokine signaling, autophagy, and host-pathogen dynamics in alveolar macrophages (AMs). While the study provides intriguing data on Tax1bp1’s involvement in Mtb growth and host inflammatory responses, significant methodological and interpretative issues undermine the robustness of its conclusions. Key concerns include inconsistent experimental conditions, lack of clarity regarding macrophage isolation and characterization, contradictions in transcriptional and functional data, and unsupported claims regarding inflammatory pathways and cell death mechanisms. The manuscript also suffers from restricted data access, limiting independent verification of the findings. Below, I provide detailed feedback and recommendations to address these issues and increase the scientific rigor of the study.

Reviewer #3: In the proposed manuscript shared by Chin et al., the authors study the role of the autophagic receptor TAX1BP1 in Mtb replication in alveolar macrophages, using in vivo and ex vivo approaches. This is a follow up study from the same group which highlighted the restriction of Mtb by TAX1BP1 in BMDMs (Budzik et al. 2020). Contrary to this report, they show a different response in alveolar macrophages (Ams), in which TAX1BP1 seems to act a pro-bacterial factor.

The authors show in a TB mouse model, the knock-out of tax1bp1 leads to decreased Mtb replication as well as decreased inflammatory cytokines production. They showed that this is also conserved in the model of Listeria monocytogenes infection. Then, by isolating specifically different cell types from mouse lungs, they described that tax1bp1 KO led to decreased Mtb burden in AMs, PMNs and MNC2, but not in MNC1. Ex vivo experiments in AMs coming from BALs provided confirmation in the pro-bacterial effect of TAX1BP1 as well as additional information. Indeed, the results highlighted that tax1bp1 KO prevents Mtb-driven necrotic-like cell death and led to earlier apoptosis. Finally, using overexpression of TAX1BP1, wt or a mutant that prevent its phosphorylation, the authors described that TAX1BP1 pro-Mtb effect is dependent on its phosphorylation.

**Part II – Major Issues: Key Experiments Required for Acceptance**

Reviewer #1: - It is interesting that the results were reproduced in two different universities (UC SF and UC Berkeley) and that the infectious strains were varied (H37Rv and Erdman), which strengthens the reproducibility and robustness of the findings. That said, the presentation is at times confusing. Experiments are sometimes analyzed with different readouts (e.g., in Figure 4, CFU for Erdman and luciferase for H37Rv). Another example is the mouse kinetics: 11, 21, and 50 days in Figure 1, while the experiment was reproduced once in Supplemental Figure 1 with kinetics at 9, 21, and 50 days. This lack of consistent timing can be confusing. Additionally, the sex of the animals is reported in Figure 1A but not mentioned elsewhere. Is there a sex effect on the susceptibility results in mice? Can you pool male and female results across all in vivo experiments and show sex differences for each group? Along the same lines, could you specify the sex of the mice from which AMs and BMDMs were extracted? These inconsistencies create an impression of poor study design. It is recommended to address these points accordingly.

- In Figure 1, a survival curve is shown. Would it be possible to include the corresponding weight curve? Furthermore, I do not understand why CFU results at 50 dpi are presented in Figure 1B when WT mice appear to have succumbed to the infection by this time (Fig 1E).

- Given the differences in cytokine production and animal survival, it is surprising to observe no changes in cellular infiltration. You have used histology to quantify this, but this technique only detects very significant changes. Could you refine this analysis using flow cytometry to quantify cellular infiltration (e.g., neutrophils) in the lungs? A deeper analysis of the results presented in Supplemental Figure 3.1 might address this question.

- Is it possible to show the weight and survival curves for the Listeria infection model? I also suggest emphasizing the relevance of testing this model in your study.

- It is known that Tax1bp1 is involved in the autophagy process. However, the colocalization results of LC3 observed very early after Mtb infection suggest that early vacuole maturation could be affected. Is it possible to verify its integrity (phagosomal rupture?) and also track the acidification of the Mtb-containing vacuole in WT and KO macrophages (in line with the results of Figure 5)?

- Regarding the reversed phenotype observed between AMs and BMDMs, is it possible to determine the expression level of Tax1bp1 over time? This could provide insights into how Mtb manipulates it. Additionally, could you reproduce experiments with one of the Mtb mutants identified in the dual RNAseq analysis?

- Regarding RNAseq analyses, have you listed genes differentially expressed between WT and KO macrophages in the uninfected state? Are macrophages predisposed or not?

- Concerning the results showing that WT macrophages undergo more necrosis than KO macrophages, is it possible to demonstrate this in the lungs of infected mice? Could you add a specific staining to your histological sections and quantify it? Alternatively, could flow cytometry be used to assess this?

- To confirm the hypothesis proposed in line 370, could you compare Mtb behavior in bone marrow-derived cells cultured with MCSF (BMDM) or GMSCF (BMDC)?

- If the conclusion is that Tax1bp1 facilitates infection in AMs but restricts it in BMDMs, could you test WT and KO mice with aerosol versus intraperitoneal (IP) infection? In this case, you might observe greater susceptibility in the lungs of WT mice following aerosol infection (as shown in Figure 1) and potentially greater resistance in the spleen and liver following IV infection.

Reviewer #2: In the first paragraph, the authors state: "Our results suggest that Tax1bp1 amplifies host-detrimental inflammatory responses, which predominate over cell-intrinsic control of Mtb replication by autophagy in macrophages (48) and contribute to host susceptibility and mortality." However, this conclusion is not supported by the data presented. The paragraph does not include specific analyses of macrophages, and the authors themselves note: "Although Tax1bp1 contributes to inflammatory cytokine synthesis during Mtb infection, microscopic examination of infected lung tissue did not reveal any significant differences in the cellular infiltrate of the lungs as reflected by lesion severity or tissue necrosis." This statement directly contradicts their assertion that host-detrimental inflammatory processes dominate over autophagy.

Based on the data presented, the observed differences in cytokine levels are more likely due to systemic inflammatory regulation rather than localized macrophage dysfunction or tissue damage. Furthermore, the statistically significant differences in cytokine levels are observed only at 21 and 50 days post-infection, corresponding to the time when monocyte-derived cells migrate to the lung parenchyma and become infected (as described in studies by Cohen et al. and Huang et al.). This strongly supports the idea that the inflammatory regulation is systemic in nature.

To strengthen the study, the authors should provide an analysis of the proportions of CD4/CD8 T-cells between WT and KO mice at early and late time points. This is especially important given that this is a whole-animal knockout model, not a myeloid-specific knockout. The deletion of Tax1bp1 is therefore likely to have broader impacts on T-cell function and could be a key factor influencing the observed cytokine differences.

The transcriptional analysis presented in the manuscript raises significant concerns. First, how do the authors reconcile their earlier finding that Tax1bp1 promotes Mtb growth in alveolar macrophages (AMs) with the observation that there are no significant transcriptional differences in Mtb between Tax1bp1 WT and KO AMs? If Mtb growth is enhanced in WT AMs but restricted in KO AMs, one would expect transcriptional changes in the bacteria reflecting this difference in growth conditions. The absence of such differences is a critical inconsistency, raising the possibility that the observed host effects are driven by systemic immune responses rather than the direct function of Tax1bp1 in AMs.

The statement, "These results are consistent with our observation that Tax1bp1 enhances increased intracellular Mtb growth during AM infection (Figure 4)," should be removed. The authors selectively highlight a few bacterial genes whose expression differences between WT and KO AMs are not statistically significant, yet claim these results support their conclusions. Such cherry-picking is misleading and does not substantiate their argument.

Regarding the host transcriptional analysis, the authors list several genes (Cd4, Cxcl1, Pf4, Pdgfa, Kitl, and Cxcl3) that they speculate are correlated with increased necrosis in WT AMs. However, no references or evidence are provided to support these claims. Moreover, the authors suggest that necrosis in WT AMs results from pro-inflammatory responses, yet their data in Figure 6 show that the intrinsic apoptotic pathway is the most upregulated in WT AMs. Adding to the confusion, they also state that pro-apoptotic genes are among the top upregulated genes in KO AMs. This contradiction undermines their conclusion that Tax1bp1 differentially regulates necrotic and apoptotic pathways.

In the Discussion section, the authors claim that "Tax1bp1 controls the mode of host cell death by initially promoting necrotic-like cell death in the first four days of Mtb infection in AMs but not BMDMs, while delaying apoptosis in the later stages of infection." However, the presented data do not support this statement and fail to demonstrate clear differences in cell death modes over time.

Finally, the GEO repository containing the transcriptional data is locked until 2028, and no password has been provided for access. This prevents reviewers from verifying the analysis. To ensure transparency and reproducibility, the authors must provide processed data files (e.g., gene count matrices) and the code used for the analysis (e.g., R scripts), allowing reviewers to validate their findings.

In the paragraph titled "Tax1bp1 promotes inflammatory cytokine signaling during AM infection ex vivo," how do the authors reconcile their findings with previous studies, such as the work by Rotchild et al., which demonstrated that AMs are unable to mount a pro-inflammatory response during the first 10 days of Mtb infection? This discrepancy is not addressed in the manuscript.

Additionally, it is not clear from the text or the Materials and Methods section at what time point the AMs were isolated. Were they harvested from naive mice, or from infected mice? If the latter, what was the time point post-infection? This information is critical for interpreting the observed inflammatory responses.

Another methodological concern is that AMs were isolated via bronchoalveolar lavage (BAL) without any further purification. It is well-established that BAL contains a heterogeneous population of cells, including tissue-resident alveolar macrophages, monocyte-derived alveolar macrophages (MoAMs), T-cells, and epithelial cells. Recent work by Russell and colleagues has shown that MoAMs are highly pro-inflammatory upon Mtb infection, in contrast to tissue-resident macrophages, which are unable to mount such responses. Thus, the observed inflammatory responses are likely driven by MoAMs, which behave similarly to bone marrow-derived macrophages (BMDMs).

To address this issue, the authors should repeat these experiments using SiglecF-based purification to isolate tissue-resident AMs, excluding MoAMs and other contaminating cell types such as T-cells. This refinement is particularly important for subsequent necrotic and apoptotic readouts, as these responses may also be confounded by the inclusion of MoAMs and other cell populations.

However, the most important issue of this study, which leads me to recommend rejection of the manuscript, is the lack of standardization and consistency in the ex vivo and in vitro experiments. According to the Materials and Methods section, infections of AMs were performed using a range of MOIs: MOI 1 in some experiments, MOI 2 in others, and MOI 10 in yet others.

Infection of macrophages at varying MOIs induces different levels of host cell activation and results in distinct readouts. For instance, infecting a monolayer at MOI 1 is fundamentally different in its cellular and immunological impact compared to MOI 10, generating data that are not directly comparable.

The majority of the results in this manuscript rely on ex vivo infections of AM monolayers, and these experiments are subsequently used to build a narrative and support the overarching conclusions. However, the inconsistent infection conditions raise concerns about the scientific validity of the study. This variability gives the impression that the experimental parameters may have been adjusted to align with the desired narrative, which undermines confidence in the findings.

For this study to be scientifically robust and relevant, the infection conditions should be standardized. All experiments should utilize consistent infection protocols, ensuring that the results are comparable and provide a reliable basis for the conclusions drawn.

Reviewer #3: This work describes macrophage type specific functions in the context of Mtb infection. However, I have several concerns that the authors need to address:

Major points

• Despite efforts made using both relevant in vivo and ex-vivo experiments, I am concerned about the number of biological replicates provided in this manuscript. At this stage of submission, many of the experiments are only performed once or twice by biological replicate, especially key experiments supporting the conclusions. In Figure 1 and Supplemental Figure 1 describes two similar experiments done in 5 mice each time, but in a single set of experiments. An additional third experiment should be important to provide clear evidence about the role of TAX1BP1 in vivo.

• In Figure 4, AMs isolation in BALs is described in the material and methods section. However, the authors need to provide characterization evidence that those cells are indeed AMs (flow cytometry, immunofluorescence for example). This characterisation is important for the conclusions.

• How these cells compared with the Siglec+ and Siglec- cells from Pisu et al. 2021?

• Related to the figure 5: On one hand, LC3 puncta, as reviewed in Kumar et al. 2021, should not be considered as a marker of autophagosomes. To confirm that those are autophagosomes, the authors should either provide TEM evidence or genetic evidence (ATG7 vs ATG13 KO for example). On the other hand, autophagosome maturation, which is the fusion of those structures with lysosomal compartments is not shown. Ideally, the authors could use the tandem RFP-GFP LC3 construct to provide such information. Alternatively, the authors could include this figure in the supplemental section and describe that they observed similar phenotypes between BMDMs and AMs leading them to further characterize this role of TAX1BP1 with different approaches.

• There is an important connection here between nitric oxide, cell death and macrophage control of Mtb. It will be important here to determine NO release from both BMDM and AM to define if autophagy independent, NO-dependent mechanisms are the underlying reason for the observed differences

**Part III – Minor Issues: Editorial and Data Presentation Modifications**

Reviewer #1: (No Response)

Reviewer #2: (No Response)

Reviewer #3: In the entire manuscript, individual values should be plotted in bar charts. In the main text, the authors systematically write the effect of tax1bp1 KO as “TAX1BP1 is acting… “. The available data, which are comparing wt and tax1bp1 KO mice cannot provide evidence, unless the authors find a model where they rescue tax1bp1 expression. The authors should reword their findings such as “tax1bp1 KO led to decreased Mtb replication suggesting that TAX1BP1 plays a pro-bacterial role”

In Figure 1, the authors should harmonize their representation as they separate male and female mice only in the first timepoint, without providing explanation in the text. Interestingly, it seems that the uptake of Mtb between male and female in tax1bp1 KO is different. Could the authors comment on this?

Figure 1 Supplemental 1A, it could be a coincidence, but the p-values at 21 and 50 days are similar to those shown in Fig1B. Could the authors double-check to make sure if those values were not duplicated?

Figure1 Supplemental 3, the authors could include a sentence about the ubiquitin signal increase shown in tax1bp1 KO as it is something shown in the brain (Sarraf et al. 2020).

Figure 3 Supplemental 2, It is unclear why the authors did not provide the CFU count for this experiment.

In Figure 5, the conclusion of this figure is not in accordance with the presented data. LC3 puncta colocalization with Mtb (as well as ubiquitin) is not a marker of autophagosome maturation. The quality of the imaging is poor, and it is unclear how the quantifications were done as the images are of poor quality.

In Figure 7, The data shown in Fig7 highlight a decrease of pro-inflammatory cytokines upon tax1bp1 KO in a timepoint where Mtb replication is not affected. The authors should provide evidence, in later timepoint and in the same experiment, that tax1bp1 KO decrease Mtb replication consistent with their previous findings.

In Figure 8, the images provided show clearly that the number of cells is not homogeneous. Hence, to be more consistent, the quantification of PI+ cells and CellEvent+ cells should be normalized by the total number of cells (quantifiable using the brightfield channel). The authors should also provide a control without infection to ensure that tax1bp1 KO does not affect cell death in their conditions.

PLOS authors have the option to publish the peer review history of their article (what does this mean? ). If published, this will include your full peer review and any attached files.

**Do you want your identity to be public for this peer review?** For information about this choice, including consent withdrawal, please see our Privacy Policy .

Reviewer #1: **Yes: ** Machelart Arnaud

Reviewer #2: No

Reviewer #3: No

**Figure resubmission:**

**Reproducibility:**



---

## [Decision Letter · Decision Letter 1]

24 Jul 2025

Tax1bp1 enhances bacterial virulence and promotes inflammatory responses during Mycobacterium tuberculosis infection

PLOS Pathogens

Dear Dr. Budzik,

Thank you for submitting your manuscript to PLOS Pathogens. After careful consideration, we feel that it has merit but does not fully meet PLOS Pathogens's publication criteria as it currently stands. Therefore, we invite you to submit a revised version of the manuscript that addresses the points raised during the review process.

Please submit your revised manuscript within 60 days Sep 22 2025 11:59PM. If you will need more time than this to complete your revisions, please reply to this message or contact the journal office at plospathogens@plos.org. Please include the following items when submitting your revised manuscript:

We look forward to receiving your revised manuscript.

Kind regards,

Pierre Santucci

Academic Editor

PLOS Pathogens

Helena Boshoff

Section Editor

PLOS Pathogens

Editor-in-Chief

PLOS Pathogens

orcid.org/0000-0003-2946-9497

Editor-in-Chief

PLOS Pathogens

orcid.org/0000-0002-7699-2064

**Additional Editor Comments:**

Dear Dr. Budzik,

Thank you for submitting your revised manuscript PPATHOGENS-D-24-02731R1 entitled “Tax1bp1 enhances bacterial virulence and promotes inflammatory responses during Mycobacterium tuberculosis infection” to PLOS Pathogens.

Sincere apologies for the slight delay during the evaluation process, as one of the reviewers from the previous round of reviewing has unfortunately declined our invitation, and thus we had to seek for a third independent expert to evaluate the revised version of your manuscript.

Your manuscript has now been carefully assessed by three experts, for which their evaluations appear below. As you will see, the three of them univocally state that this revised version has significantly improved. However, two out of three experts still believe that your manuscript under its current form is not suitable for publication in PLOS Pathogens.

After discussion within the editorial board, we would be happy to consider a revised version of the manuscript as we believe that most of their comments can be addressed without an extensive experimental revision.

Therefore, we would like to invite you to submit a revised version that takes into account all the comments and suggestions raised by the experts during this second round of review.

Please submit alongside the revised version of your manuscript, a point-by-point response letter that address each concern raised by the referees.

The revised version alongside the two point-by-point responses provided during the revision process will be thoroughly assessed internally by the editors who will make a final decision regarding the suitability of your manuscript for publication in PLOS Pathogens.

Thank you again for submitting your work to PLOS Pathogens.

Sincerely yours,

Pierre Santucci

**Journal Requirements:**

2) We have noticed that you have uploaded Supporting Information files, but you have not included a list of legends. Please add a full list of legends for your Supporting Information files after the references list.

**Reviewers' Comments:**

Reviewer's Responses to Questions

**Part I - Summary**

Reviewer #1: In light of the elements provided by the authors in response to my suggestions, most of the concerns have been addressed, and the revised version of the manuscript is significantly improved and now suitable for publication in PLOS Pathogens.

Reviewer #3: Apologies the review took longer than expected, I realised the supplementary figures were missing in the original resubmission and that caused delays. I hope the other reviewers received the figures as well as I only got the file per email.

In their revised manuscript, the authors addressed some of the comments and concerns raised. In particular, the authors added another in vivo experiment in order to corroborate their findings regarding TAX1BP1 effect on Mtb replication in different macrophages (AMs, PMNs, MNC1 and MNC2). While I appreciated the efforts made by the authors, the data presented, as well as their analysis, are not suitable for publication.

A major issue of the revised manuscript relies on the lack of statistical analysis across different biological replicates and experiments. Importantly, for the Mtb infection analysis in different cell types, it is unclear why the authors decided to represent 3-4 independent experiments in different figures. Also, the infection times do not match. Not only it is hard to read, but it also prevents any robust validation of the findings regarding TAX1BP1 effect across several cell types, which is an important message of the paper.

Specifically, for the Zeon Green analysis, the individual experiments are shown in Figure 3, Figure 3-supp1, Figure 3-suppl2 and Figure 1-supp5. For CFU analysis, the individual experiments are spread in Figure 3, Figure 3-supp2 and Figure1-supp5. I believe this is not good practice.

In Figure 3-suppl1, the legend indicates “The p-values from t-test comparisons are shown” but the stats are not shown in the graph.

Reviewer #4: The revised manuscript is improved, but still lacks the essence of a coherent story.

The main narrative and overarching conclusions are built on ex vivo infections of AM monolayers, which shows different roles of tax1bp1 in AMs versus BMDMs. The discrepancy between in vivo and in vitro role of tax1bp1 in different macrophages is the most interesting part. But the manuscript reads like a few small loosely related phenotypes pieced together. It has distractive in vivo studies and listeria data deviating from the main story. The manuscript also suffers from an unclear definition of “inflammatory response”. Which of inflammatory responses is tax1bp1 regulating? Is it cell death? Is it AM production of IL1b, IFNg or PGE2? Or immune response after Mtb infection? Each of these “inflammatory responses” has a completely distinct meaning.

**Part II – Major Issues: Key Experiments Required for Acceptance**

Reviewer #1: (No Response)

Reviewer #3: (No Response)

Reviewer #4: 1. I do not understand why CFU results at 50 dpi are presented in Figure 1B when the mice appear to have succumbed to the infection much later, not until after 30 weeks, and the role of tax1bp1 in survival doesn’t appear even later. This discoordination between CFU/cell death/mortality suggest that the defect in controlling bacterial replication does not contribute directly to host mortality. This needs to be carefully discussed.

2. Figure 1D. Tax1bp1 deficient mice has lower IFNg and other cytokines in the infected lungs. For the record, IFNg is produced by T cells. Yet the authors found increased CD8 T cell levels. How does decreased IFNg reconcile with increased T cells?

3. Figure 2E is poorly labeled, please include time point in the figures. For this whole figure, I don’t understand the role of Listeria data. The only common thing between listeria and Mtb might be that, after Listeria infection there is a reduced production of multiple cytokines, several of which involved in controlling Listeria. But this has nothing to do with AM or may not have to do with any type of macrophages. what’s the point of including Listeria? The authors stated this part is to “confirming the importance of our findings with another intracellular pathogen”. what findings are they referring to?

4. In Figure 3-S2B, tax1bp1 deficiency results in decreased bacterial burden in AM, but increased CFU in PMN and MNC1, at 21 dpi. Why? Since the CFU results obtained from PMN and MNC1 is different between before and after 21dpi, why is the overall lung burden shows similar reduction in Tax1bp1 deficient mice at 7, 11, 14, 21, and 50 dpi?

5. Figure 5 and 6. The whole section on RNA-seq leads to no meaningful conclusion. I have the same question as one other reviewer: intrinsic apoptotic pathway is the most upregulated in WT AMs. Adding to the confusion, they also state that pro-apoptotic genes are among the top upregulated genes in KO AMs. This contradiction undermines their conclusion that Tax1bp1 differentially regulates necrotic and apoptotic pathways. Adding on that, if, as the authors stated in response letter, complexity of signalling pathways limits the predicting power of RNA-seq data, why having it in the manuscript?

6. in Figure 7, what are the authors trying to do with cytokine secretion from AMs? Are the authors trying to link observation in Figure 1, IL1b level differs at 11dpi from Mtb infected mice? However, IL1b is higher in tax1bp1 deficient mice, contradicting to figure 7. If the authors believe tax1bp1 regulation of PGE2 is the cause, addition experiment is needed to solidify this point.

7. LC3 puncta is not a marker of autophagosome maturation. Does Tax1bp1 affect autophagosome formation? The authors at least need to provide western blot for both AM and BMDM. Does tax1bp1 KO affect cell death without Mtb infection?

8. Autophagy is required for controlling Mtb growth in vivo and ex vivo is less accurate as bactericidal activity of autophagy is still under debate (two important reference missed in the manuscript. PMID: 38413834, 39242815). Please at least rephrase as “Autophagy is required for controlling Mtb in vivo and ex vivo”.

9. It is very annoying to read when the authors interpreting the data as the function of tax1bp1 rather than stating the phenotype of tax1bp1 deficiency in most places. Such statements also imply for gain-of-function studies, rather than genetic LOF, which deviate from the facts. I agree with reviewer 3 and suggest rewording. I disagree with the authors: the manuscript is not easier to read nor conceptualize when phrased to point out the effects of Tax1bp1.

Examples include, but not limited to, the following:

Tax1bp1 increased levels of IL-6, TNF-α, IL1-β, and IL-12/IL-23 p40

Line 228: At 48 hours, Tax1bp1 significantly increased IL-6,…

Line 232: Indeed, Tax1bp1 increased CFU..

Line 233: As with the IV infection, Tax1bp1 led to a considerable non-statistically…

Line 290: Tax1bp1 slightly decreased Mtb colocalization..

Line 347: Tax1bp1 upregulated two Mtb genes, mmpL4 and mbtE…

Line 387: Tax1bp1 led to increased levels of several inflammatory cytokines..

**Part III – Minor Issues: Editorial and Data Presentation Modifications**

Reviewer #1: (No Response)

Reviewer #3: (No Response)

Reviewer #4: Line 164 grammar error: Therefore, we measured interferon levels using a type I and II IFN reporter cell line (ISRE) and a type II IFN by ELISA.

PLOS authors have the option to publish the peer review history of their article (what does this mean? ). If published, this will include your full peer review and any attached files.

**Do you want your identity to be public for this peer review?** For information about this choice, including consent withdrawal, please see our Privacy Policy .

Reviewer #1: **Yes: ** Arnaud Machelart

Reviewer #3: No

Reviewer #4: **Yes: ** Yating Wang

**Figure resubmission:**

**Reproducibility:**



---

## [Editor Report · Decision Letter 2]

6 Oct 2025

Dear Budzik,

We are pleased to inform you that your manuscript 'Mycobacterium tuberculosis triggers reduced inflammatory cytokine responses and virulence in mice lacking Tax1bp1' has been provisionally accepted for publication in PLOS Pathogens.

Best regards,

Pierre Santucci

Academic Editor

PLOS Pathogens

Helena Boshoff

Section Editor

PLOS Pathogens

Sumita Bhaduri-McIntosh

Editor-in-Chief

PLOS Pathogens

orcid.org/0000-0003-2946-9497

Michael Malim

Editor-in-Chief

PLOS Pathogens

orcid.org/0000-0002-7699-2064

Dear Dr. Budzik,

Thank you again for choosing PLOS Pathogens for your study. The editorial team has now carefully inspected the latest version of your manuscript in addition to your responses to the referees' comments.

The editorial team thinks that the authors have addressed the reviewers’ comments and revised their manuscript accordingly. The current version meets the standards of PLOS Pathogens and is suitable for publication.

Thank you again for your investment in these thoughtful and comprehensive revisions.

Sincerly yours,

Pierre SANTUCCI - PLOS Pathogens Academic Editor.
---

## [Editor Report · Acceptance letter]

Dear Budzik,

We are delighted to inform you that your manuscript, "Mycobacterium tuberculosis triggers reduced inflammatory cytokine responses and virulence in mice lacking Tax1bp1," has been formally accepted for publication in PLOS Pathogens.

Best regards,

Sumita Bhaduri-McIntosh

Editor-in-Chief

PLOS Pathogens

orcid.org/0000-0003-2946-9497

Michael Malim

Editor-in-Chief

PLOS Pathogens

orcid.org/0000-0002-7699-2064